# Ultrahigh pressure compaction-resistant thin film crosslinked composite reverse osmosis membranes

Jishan Wu[1,2,11], Javier A. Quezada-Renteria[1,11], Jinlong He [3], Minhao Xiao[1], Yuanmiaoliang Chen[2], Hanqing Fan [2], Xinyi Wang [1], Fiona Chen[4], Kevin Pataroque [5], Yara Suleiman[6], Sina Shahbazmohamadi[6], N. A. Sreejith[7], Hariswaran Sitaraman[7], Marc Day [7], Ying Li[8], David Jassby [1], Jeffrey R. McCutcheon [9], Menachem Elimelech [2,4] & Eric M. V. Hoek [1,10] ✉

In this study, we present a class of thin-film crosslinked (TFX) composite reverse osmosis (RO) membranes that resist physical compaction at ultrahigh pressures (up to 200 bar). Since RO membranes experience compaction at virtually all pressure ranges, the ability to resist compaction has widespread implications for RO membrane technology. The process described herein involves crosslinking a phase inverted porous polyimide (PI) support membrane followed by interfacial polymerization of a polyamide layer, thereby forming a fully thermoset composite membrane structure. We explore a range of phase inversion membrane formation parameters such as PI concentration, solvent-cosolvent ratios, coagulation bath composition, and crosslinking methods in addition to interfacial polymerization reaction chemistry and conditions. Overall, TFX membranes exhibit significantly less compaction compared to hand-cast and commercial high-pressure RO membranes, experiencing less than 10% decline in water permeance and maintaining salt rejection over 99% for NaCl solutions up to 180,000 mg/L with 200 bar applied pressure.

Reverse osmosis (RO) is a highly effective process to economically produce clean, fresh water from almost any water source[1-4]. The majority of RO membranes in commercial use are thin-film composite (TFC), composed of three layers. The thin polyamide (PA) active layer is formed directly atop a mesoporous support layer, which itself is cast through phase inversion onto a macroporous nonwoven fabric substrate[5]. The PA layer is normally synthesized via a polycondensation reaction between multi-functional amine and acyl halide monomers. *Meta*-phenylenediamine (MPD) and trimesoyl chloride (TMC) are most commonly employed, although various other monomers and their combinations have been explored to fine-tune membrane performance for specific RO and nanofiltration (NF) needs.

Modern PA-PSU-PET composite RO membranes excel in desalination and advanced water treatment due to their excellent

[1]Department of Civil & Environmental Engineering, University of California, Los Angeles, CA, USA. [2]Department of Civil & Environmental Engineering, Rice University, Houston, TX, USA. [3]Failure Mechanics and Engineering Disaster Prevention Key Laboratory of Sichuan Province, Sichuan University, Chengdu, China. [4]Department of Chemical & Biomolecular Engineering, Rice University, Houston, TX, USA. [5]Department of Chemical & Environmental Engineering, Yale University, New Haven, CT, USA. [6]Department of Biomedical Engineering, University of Connecticut, Storrs, CT, USA. [7]Computational Science Center, National Renewable Energy Laboratory, Golden, CO, USA. [8]Department of Mechanical Engineering, University of Wisconsin-Madison, Madison, WI, USA. [9]Department of Chemical & Biomolecular Engineering, University of Connecticut, Storrs, CT, USA. [10]Energy Storage & Distributed Resources Division, Lawrence Berkeley National Lab, Berkeley, CA, USA. [11]These authors contributed equally: Jishan Wu, Javier A. Quezada-Renteria. ✉e-mail: emvhoek@ucla.edu

combination of water permeance and salt rejection. In nearly all applications, TFC membranes lose a fraction of their initial permeance due to densification of the PSU layer. Thermoplastics like PSU are generally susceptible to "plastic creep" and permanent deformation when exposed to high pressures and temperatures, but are referred to as "compaction" in the field of membrane science[5–13]. Membrane compaction can result in significant permeability loss year-over-year, even at fairly low applied pressures[14]. RO membrane compaction is largely irreversible, although recent studies suggest it may also happen to the PA layer at extreme applied pressures, but is reversible due to the high crosslinking degree of the PA layer[5,9].

Developing compaction-resistant membranes is particularly important for high-pressure and ultra-high-pressure RO applications such as seawater desalination, direct brine concentration, brine concentration via osmosis-assisted multistage RO processes, and various resource recovery operations[15–17]. However, even lower-pressure NF and RO membranes could benefit from enhanced compaction resistance—including in low-pressure brackish water RO, ultralow-pressure wastewater RO, and offshore sulfate removal NF membranes—by improving physical stability and service lifespan[18]. While thermoset polyamide coating films compact only to a small extent, they recover back to their starting configuration upon release of the applied pressure; this is specifically due to and directly proportional to the degree of crosslinking[5]. In contrast, the irreversible compaction of the PSU support layer happens because it is a thermoplastic[5].

Herein, we use a commercial polyimide (PI) material to serve as an exemplar polymer that has excellent intrinsic mechanical and chemical properties, and which is easily crosslinked[19,20]. Crosslinked PI membranes were initially developed for gas separations to reduce plasticization[21] and later applied successfully to pervaporation[22,23] and organic solvent-resistant NF membranes[24–29]. Various crosslinking methods for PI have been employed, including thermal[30], UV[21], and chemical crosslinking[31,32]. Livingston et al. reported the post-synthesis crosslinking of PI membranes with aliphatic diamines (ethylene diamine, propylene diamine, 1,6-hexandiamine, and 1,8-octanediamine), and the crosslinked membranes exhibited almost no compaction after 120 h testing at 30 bar applied pressure[33]. Herein, we refer to this post-synthesis crosslinking as "ex situ" crosslinking, and we introduce an "in situ" method – described in detail below. A few previous studies used crosslinked PI as the support layer for composite RO membranes, but only applied 0.6–30 bar feed pressure, and the salt rejections were fairly low (39%–97%)[28,34].

In this study, we introduce a systematic approach to produce ultra-high pressure-tolerant RO membranes with excellent separation performance and compaction resistance—defined as the membrane's ability to retain water permeance and salt rejection at ultra-high applied pressure. This process involved converting a PI-based porous support membrane into a thermoset via crosslinking with an aliphatic diamine, followed by the formation of a polyamide active layer via interfacial polymerization. Systematic optimization of polymer concentration, solvent/co-solvent ratios, phase inversion bath composition, and interfacial polymerization conditions enabled tailoring both the support membrane and the coating film structure, morphology, and performance. We present a range of crosslinked composite RO membranes that resist physical compaction and the associated loss of water permeability and salt rejection for salt (NaCl) concentrations up to 180,000 mg/L and feed pressures up to 200 bar.

## Results
### Formation, properties, and separation performance of crosslinked porous support membranes
The formation processes, physical-chemical properties, and separation performances shown in Fig. 1 demonstrate substantial improvements in uncoated, crosslinked PI porous support membrane mechanical strength, compaction resistance, and a range of different separation performance properties. Figure 1a depicts the in situ and ex situ phase inversion and crosslinking processes. The ex situ crosslinking involves phase inversion followed by crosslinking, whereas in the in situ approach, the PI was directly phase inverted in the crosslinking solution, so both phase inversion and crosslinking occur simultaneously. This simplifies the fabrication process and makes it more practical for scaling up to commercial manufacturing.

The uncrosslinked PI membrane exhibits a higher tensile strength than a PSU porous support membrane; the latter is the most commonly used polymer for RO support membranes (Fig. 1b). A recent study by Lim et al.[8] suggests that a support layer with a tensile strength higher than 7 MPa is needed to fabricate mechanically robust membranes for HPRO applications. Both ex situ and in situ crosslinking (Fig. 1a) successfully increased tensile strength to 14-19 MPa which is about 4-5 times higher than the standard PSU support.

FTIR spectra (Fig. 1c) provide clear evidence of chemical crosslinking turning the PI layer into a thermoset material, as indicated by the shift from imide to amide functional groups in both ex situ and in situ PI membranes. Characteristic imide absorption bands, highlighted in red, appear at 1780 and 1711 cm$^{-1}$ for the carbonyl (C=O) stretching and at 1361 cm$^{-1}$ for the carbon-nitrogen (C–N) stretching. In contrast, the amide bands, marked in blue, are identified at 1634 cm$^{-1}$ for the carbonyl stretching and at 1525 cm$^{-1}$ for the C–N stretching, indicative of amide linkages that result from crosslinking.

Water flux data (Fig. 1d) shows that the tested porous support membranes exhibit varying degrees of compaction at one bar applied pressure. PSU membranes undergo the most flux decline due to compaction, losing nearly half of their initial permeance after one hour, whereas PI membranes are slightly more compaction-resistant due to PI's superior mechanical strength versus PSU. Both ex situ and in situ crosslinked PI membranes show even better compaction resistance with only about 10 percent flux decline – both stabilizing around 550 LMH.

A previous study investigated the effects of support membrane properties such as pore size, surface, and body porosity on the PA film mass, thickness, crosslinking degree, and separation performance of the PA-PSU composite RO membranes[35]. Herein, we explore a multitude of variables (i.e., polymer concentration, solvent and co-solvent ratio, coagulation bath chemistry, and crosslinking approaches) to fine-tune the support membrane structure and morphology and to make the support suitable for interfacial polymerization. The scanning electron microscope (SEM) images of membrane surfaces are displayed in Supplementary Fig. 1. The calculated porosity and contact angles are provided in Supplementary Table 1. As the polymer concentration in the casting solution increases, the resulting support membrane becomes denser due to the decreasing surface porosity, which is attributed to slower phase inversion kinetics. This reduction in porosity leads to increased hydrophilicity, as fewer air-filled voids reduce water contact angles[36].

Supplementary Figs. 2–6 illustrate differences in PI support membrane surface porosities and macrovoid morphologies based on changes to coagulation bath composition, solvent-to-co-solvent ratios, and crosslinking. Switching the coagulation bath from DI water to IPA results in a denser membrane; IPA has a lower affinity for the organic solvent, which slows down the phase inversion process, producing smaller and fewer surface pores. Similarly, higher co-solvent ratios also resulted in denser, less porous membranes due to slower phase inversion. Both ex situ and in situ crosslinking reduced surface porosity by promoting polymer network densification near the surface, as shown in Supplementary Fig. S3. This effect is consistent with prior findings that crosslinking restricts chain mobility and reduces free volume, yielding tighter and less porous structures[27]. Denser interfacial layers were also observed at high polymer concentrations, coagulation using IPA instead of DI, and high co-solvent ratios (Supplementary Figs. 5i, j, and 6d–f).

 

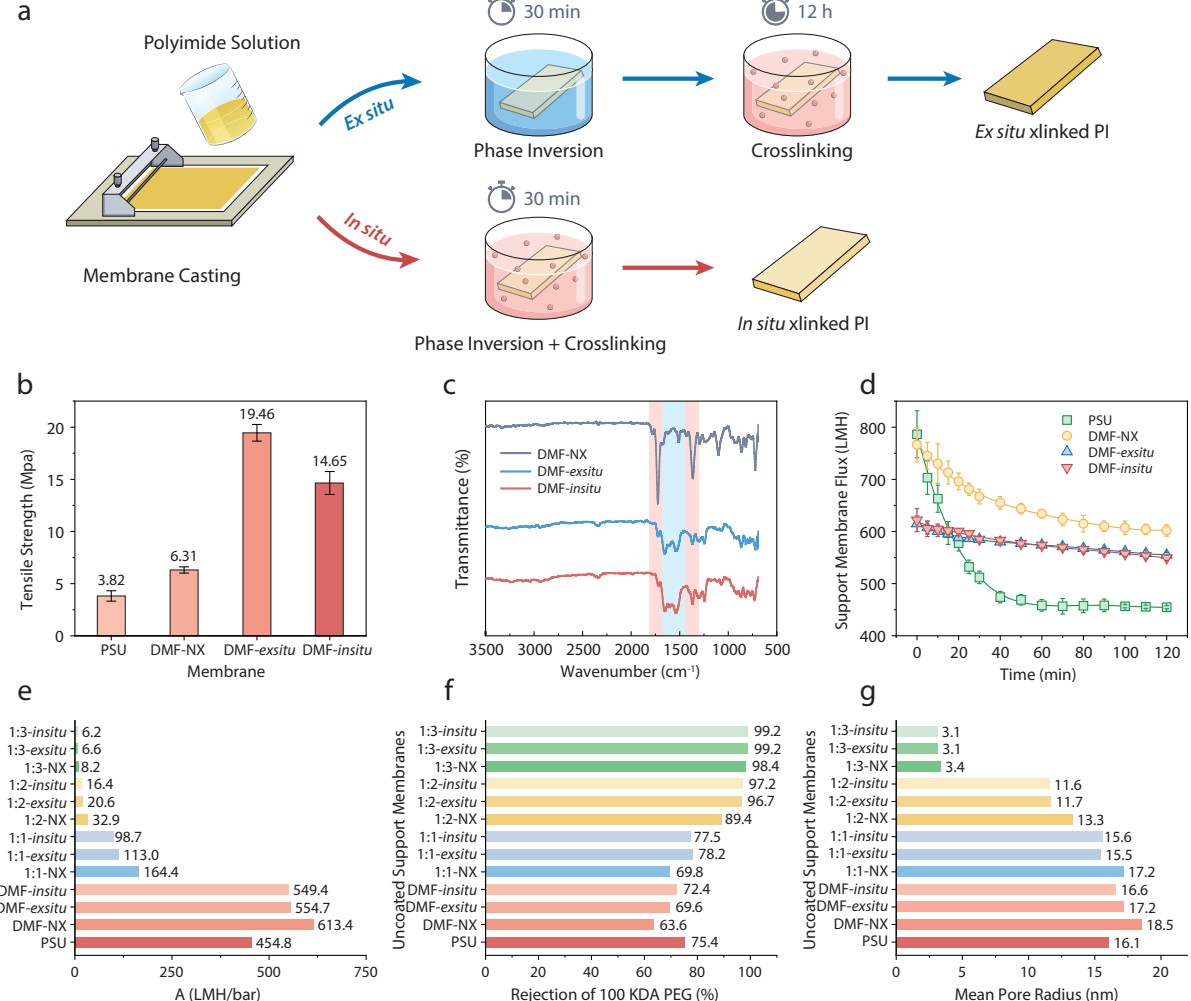

**Fig. 1 | Processes, properties, and performance of uncoated support membranes. a** Schematic illustration of ex situ and in situ crosslinking of the PI support membranes. **b** Tensile strength of PSU, non-crosslinked PI (DMF-NX), ex situ (DMF-*exsitu*), and in situ crosslinked PI (DMF-*insitu*) support membranes (18% polymer in DMF phase inverted in IPA). **c** Fourier-transform infrared (FTIR) analysis of non-crosslinked PI, ex situ and in situ crosslinked PI support membranes. **d** Fluxes of DI water through exemplary support membranes at 1 bar applied pressure. **e** DI water permeances of all uncoated support membranes. **f** Rejection of 100 kDa PEG and **g** mean pore radius of uncoated support membranes. Water flux is presented in LMH (Liter m$^{-2}$ h$^{-1}$), and water permeance is presented in LMH/bar (Liter m$^{-2}$ h$^{-1}$ bar$^{-1}$). Data represent the mean ± standard deviation from three independent experiments, where error bars are shown.

From a structural standpoint, both in situ and ex situ crosslinking reduce surface porosity by promoting polymer densification, but their crosslinking profiles may differ due to diffusion kinetics. Ex situ crosslinking typically results in more surface-concentrated modification, while in situ crosslinking enables more uniform integration as the diamine diffuses through the semi-coagulated polymer during phase inversion. This difference is evident in membranes with more porous structures, such as in the comparisons of 1:2 ex situ vs. 1:2 in situ and 1:1 ex situ vs. 1:1 in situ. In these cases, the in situ/ approach leads to more extensive densification throughout the membrane matrix, resulting in lower water permeance and slightly higher rejection. In contrast, the ex situ method primarily modifies the surface layer, maintaining rejection performance but allowing higher water permeance due to less uniform densification.

Despite similar porosity outcomes, the in situ method offers process advantages by combining phase inversion and crosslinking in a single step, reducing fabrication time and improving scalability.

Figure 1e–g shows the results of wet-testing for all the uncoated PI (and PSU) support membranes exhibiting a wide range of permeabilities (~6 to ~600 LMH/bar). All crosslinked PI membranes have slightly smaller average pore sizes than uncrosslinked PI membranes, but changes to the solvent system and coagulation bath composition drove greater differences.

**Compaction resistance of the TFX composite RO membranes**
Polyimide support membranes are generally more hydrophilic than polysulfone-based support membranes (Supplementary Table 1). As a result, the amine monomer uptake of PI supports during interfacial polymerization may differ from that of polysulfone supports. Furthermore, considering the high affinity of the diamine monomer for the imide bond in the polymer backbone, a higher than typical amine monomer concentration of 6.0% (w/w) is used to ensure adequate free amine monomer is available to participate in the interfacial polymerization reaction.

Water permeance ($A$-values), rejections ($R_{ob}$), and solute permeability ($B$-values) of ex situ formed TFX membranes are provided in Fig. 2a and Supplementary Table 2. Ex situ TFX membrane with tighter support membranes experiences less than 10% compaction; these supports utilize 16%−20% PI concentration and a 3:1 ratio of 1,4-dioxane:DMF with pure IPA as the nonsolvent. The compaction resistance of the ex situ TFX membranes stems from their fully thermoset construction, with both the PA active layer and the support layer highly crosslinked. The rejections of ex situ TFX membranes decline with increasing pressure because their salt permeance increases.

The in situ TFX membranes also exhibit strong compaction resistance, comparable to their ex situ counterparts (Fig. 2b, Supplementary Table 3). Notably, some in situ TFX membranes (18%, 20%, 1:1 co-solvent/solvent ratio) maintain high salt rejection rates of up to 99.05% at pressures reaching 200 bar. The in situ TFX membrane prepared with 20%-3:1 condition (*i-TFX-20%-3:1*), with its higher water permeance and salt rejection, demonstrates that the in situ crosslinking approach is not only a more scalable fabrication process, but also yields superior performance than the ex situ approach. The superior performance of *i-TFX* is attributed to the favorable match between its support structure and the chosen interfacial polymerization conditions.

Although all TFX membranes were coated using the same formulation, the resulting active layer morphologies differed (Fig. S7), reconfirming how subtle differences in support properties can affect monomer uptake and polyamide layer development[35]. For in situ crosslinking, the crosslinker is present during phase inversion, allowing it to diffuse into the nascent, semi-coagulated polymer matrix. This facilitates more uniform densification throughout the support membrane's near-surface layer, reducing both surface and body porosity. As established in a previous study[35], reduced bulk porosity directly limits the uptake of the aqueous MPD monomer, which subsequently lowers the local amine concentration available during interfacial polymerization. This leads to the formation of thinner PA layers with reduced mass and crosslinking degree, thus increasing water permeance, but potentially reducing rejection.

In contrast, ex situ crosslinking occurs after the support membrane has already formed. The crosslinking reaction primarily affects the surface region. As a result, ex situ treated supports typically retain higher body porosity, enabling greater MPD uptake, especially near the top surface. This increased monomer availability translates into thicker and more crosslinked PA layers, leading to lower water permeance and higher salt rejection. Therefore, the variation in PA layer structure between in situ and ex situ crosslinked supports arises from differences in both surface and bulk porosity, which control the mass of monomer sequestered within the support. This, in turn, determines the MPD:TMC ratio in the interfacial reaction zone and governs the final PA layer morphology and performance. The low permeance of these TFX membranes is a result of their tight crosslinked PI support layers (<10 LMH/bar vs. ~450 LMH/bar for PSU supports). Enhancing the permeance of the TFX composite RO membranes could be achieved by increasing the support layer surface porosity.

In Fig. 2c, water permeance at 200 bar ($A_{200}$) and the ratio of permeance at 200 bar to that at 60 bar ($A_{200}/A_{60}$) serve as key indicators of membrane performance. As the support structure becomes denser (via higher polymer concentrations and increased co-solvent/solvent ratios), compaction resistance improves. When using a 1:1 co-solvent/solvent ratio, polymer concentration has minimal impact on permeance but significantly enhances compaction resistance. In contrast, a 3:1 ratio boosts both compaction resistance and water permeance for both *i-TFX* and *e-TFX* membranes compared to their 1:1 counterparts. The best-performing membranes, *i-TFX-20%-3:1 (i-TFX)* and *e-TFX-18%-3:1 (e-TFX)*, exhibit nearly 100% compaction resistance and ~99% salt rejection under all testing conditions up to 200 bar. These membranes were selected for further characterization,

optimization, and comparison against state-of-the-art thin-film composite (TFC) high-pressure RO (HPRO) and hand-cast PSU-TFC RO membranes.

Supplementary Table 4 compares non-crosslinked PI-TFC membranes with crosslinked TFX variants, illustrating distinct differences in RO performance at increasing pressures. TFC membranes with higher polymer concentrations—such as TFC 2 (18% PI) and TFC 3 (20% PI)—exhibit higher salt rejection at lower pressures, but reduced water permeability compared to TFC 1 (16% PI), highlighting the typical trade-off between selectivity and permeability. TFC 4 (16% PI, IPA-processed) shows lower salt rejection, likely due to strong, favorable interactions between the PI support and MPD during polyamide layer formation. TFC 11 (18% PI, 1:1 solvent-to-co-solvent ratio) retains high salt rejection at lower pressures, but shows a significant decline at 200 bar. The non-crosslinked membrane performance emphasizes the importance of crosslinking for compaction resistance at ultra-high pressure operation.

Supplementary Fig. S7 shows the surface morphology of PA active layers on both ex situ and in situ TFX membranes, which directly impacts their rejection capabilities. The performance of RO membranes is closely linked to the properties of their active PA layers, which are significantly influenced by the underlying support membrane's properties[35]. Membranes with smaller hemispherical structures typically indicate less densely crosslinked PA films, which could account for the low rejection rates. In contrast, ex situ TFX (*e-TFX-3:1*) exhibits pronounced ridge-and-valley structures in its PA layers. This morphology tends to correlate with higher crosslinking degrees, which produces higher salt rejection, especially at high applied pressure and salinity, as tested herein[5]. Ex situ TFX membranes with lower salt rejection (*e-TFX-1:1*) display less dense, nodular morphology, which tends to correlate with lower crosslinking degree and lower salt rejection. The leaf-like surface protuberances of the in situ TFX membranes indicate effective interfacial polymerization, forming dense ridge-and-valley structures. Variations between membranes, such as smaller hemispherical protuberances in *i-TFX-20%-3:1* versus more complex structures in *i-TFX-1:1*, suggest differences in crosslinking.

## Improving TFX membrane performance by tailoring PA interfacial polymerization

Triethylamine (TEA) acts as a buffer and neutralizes hydrochloric acid when the amine and acid chloride monomers react to form the amide bond during interfacial polymerization[37]. "Tailored TFX" (*t-TFX*) membranes have significantly higher A and lower B-values, flux, and rejection rates at elevated pressures compared to both ex situ and in situ TFX membranes. TEA, a tertiary amine, functions as a catalyst in the polymerization reaction between MPD and TMC by scavenging hydrogen halides formed during amide bond formation[37]. This catalytic action mitigates the inhibition of the polymerization process, thereby preventing defects in the PA layer[37,38]. The result is a smoother, more uniform PA selective layer, which enhances separation performance and minimizes surface defects. Furthermore, TEA influences the degree of crosslinking in the PA layer, a critical factor for controlling membrane permeability and selectivity[5,39]. A controlled reaction environment, facilitated by TEA, yields an optimally crosslinked network that improves both rejection performance and flux. Supplementary Figs. S8 and S9 show the surface SEM and AFM images of the *t-TFX*, respectively. While SEM does not reveal distinct morphological differences between *t-TFX*, *e-TFX*, and *i-TFX*, AFM analysis shows that *t-TFX* has a higher surface roughness ($R_q$ = 114.7 nm) compared to *e-TFX* ($R_q$ = 81.2 nm) and *i-TFX* ($R_q$ = 64.6 nm). This suggests a thicker PA layer formed on the *t*-TFX, which may explain its enhanced separation performance.

As shown in Fig. 3a and Supplementary Table 5, the *t-TFX* membrane exhibits ~3X the water permeance (0.73 LMH/bar) compared to

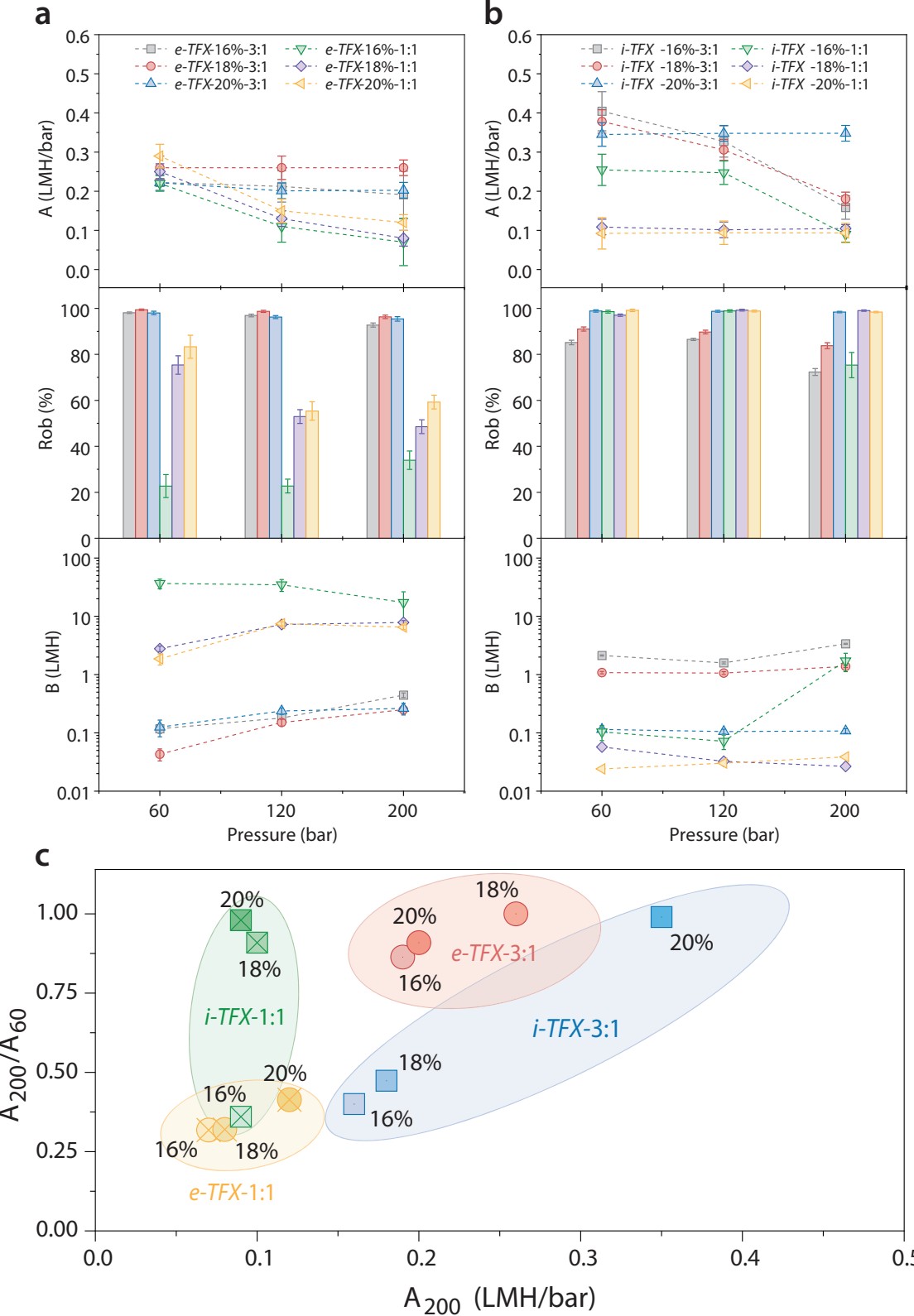

**Fig. 2 | TFX membrane wet-testing performance. a** Water permeance (*A*), observed rejection (*R*$_{ob}$), and salt permeability (*B*) of ex situ TFX (*e-TFX*) fabricated using support membranes crosslinked ex situ with varying polymer concentrations (16%–20%) and co-solvent/solvent ratios (1:1 or 3:1). **b** *A*, *R*$_{ob}$, and *B* of in situ TFX (*i-TFX*) prepared with in situ crosslinked support membranes. **c** Water permeance ratio at 200 bar versus 60 bar (*A*$_{200}$/*A*$_{60}$), plotted against the final water permeance of TFX membranes. Water permeance is presented in units of LMH/bar (Liter m$^{-2}$ h$^{-1}$ bar$^{-1}$). Solute permeability is presented in units of LMH (Liter m$^{-2}$ h$^{-1}$). Data represent the mean ± standard deviation from three independent experiments, where applicable.

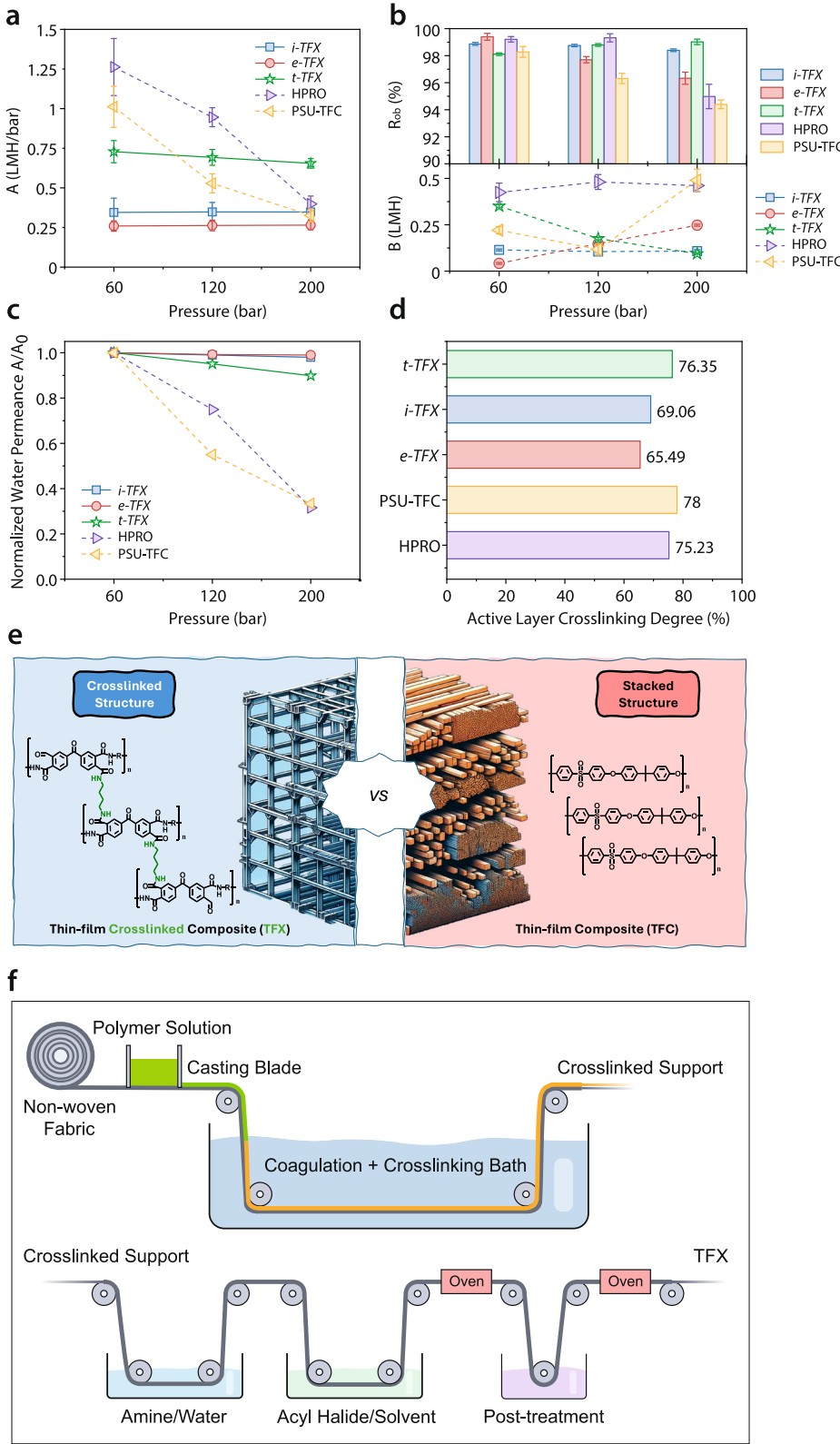

**Fig. 3 | Tuned TFX membrane wet-testing performance. a** Water permeance of *t-TFX* compared to *e-TFX*, *i-TFX*, hand-cast PSU-TFC, and commercial HPRO. **b** Observed rejection and salt permeability of t-TFX compared to other counterparts. **c** Normalized water permeance of TFX membranes versus hand-cast PSU-TFC and commercial HPRO. **d** Active layer crosslinking degree of TFX membranes, hand-cast PSU-TFC, and commercial HPRO. **e** Schematic of enhanced mechanical strength and stability of the TFX composite membrane by turning thermoplastic into thermoset. **f** Schematic illustration of adapting existing TFC membrane manufacturing lines for roll-to-roll production of TFX membranes. Water permeance is presented in LMH/bar (Liter m$^{-2}$ h$^{-1}$ bar$^{-1}$). Solute permeability is presented in LMH (Liter m$^{-2}$ h$^{-1}$). Data represent the mean ± standard deviation from three independent experiments, where applicable.

*e-TFX* (0.26 LMH/bar) and ~2X to *i-TFX* (0.35 LMH/bar) at 60 bar. Notably, the permeance of *t-TFX* decreases by less than 10% when the pressure is increased up to 200 bar (0.65 LMH/bar). Its observed rejection (Fig. 3b) improves from 98.11% ± 0.14% at 60 bar to 98.79% ± 0.23% at 120 bar, and further to 99.02% ± 0.25% at 200 bar, attributed to compacted PA active layer (decreasing *B*-value). These results indicate that *t-TFX* membranes possess superior handling capabilities and versatility across various operational scenarios. The slightly more compaction observed in *t-TFX*, in contrast to *e-TFX* and *i-TFX,* could be attributed to its higher flux (Supplementary Fig. S10)[40]. For applications such as UHPRO, the balance between flux and rejection is critical. High flux can lead to s high concentration polarization (CP) modulus, potentially compromising the process due to elevated osmotic pressures induced by high CP[15]. Therefore, the moderate flux and high rejection of *t-TFX* make it a suitable candidate membrane for UHPRO applications, facilitating sustainable brine concentration.

The hand-cast PSU-TFC membrane, prepared using standard phase inversion and IP coating procedures as reported in a previous study[35], and the commercial HPRO membrane, both lacking crosslinking on the PSU support layer and thus being thermoplastics, experience significant reductions in water permeance, with losses approaching 70% at 200 bar (Fig. 3c). At pressures below 120 bar, the commercial HPRO membrane displays higher water permeance than TFX membranes, attributable to its more porous interfacial and support layers (Fig. 1e–g). Most membrane resistance stems from the active layer and the interfacial layer in the composite membranes. Future work should prioritize enhancing the permeance of the interfacial layer of the support membranes while preserving compaction resistance[40]. Contact angle measurements indicate that *t-TFX* exhibits greater hydrophilicity than *e-TFX* and *i-TFX* (Supplementary Table S6). In addition, Fig. 3d highlights a higher crosslinking degree in *t-TFX*, as verified by XPS analysis (also Supplementary Table S6), confirming the enhanced effectiveness of TEA in improving both the interfacial polymerization process and crosslinking density within the selective layer.

The stark comparison between conventional TFC membrane and TFX membrane validates the hypothesis that crosslinking transforms the composite membrane into a thermoset, enhancing mechanical strength and compaction resistance (Fig. 3e). Unlike thermoplastic membranes, which rely on weaker intermolecular forces and are prone to deformation under high pressure, the covalent bonds formed during crosslinking in TFX membranes create a rigid, interconnected polymer network. These covalent linkages prevent structural deformation under pressure. The improved structural resilience of TFX membranes under ultrahigh pressures directly enhances their applicability for high TDS brine concentration. Compaction-resistant membranes like *t-TFX* retain over 90% of their initial permeance ($A_{200}/A_{60} > 0.9$), while maintaining over 99% salt rejection, compared to commercial HPRO membranes that suffer ~70% loss in permeance and significant rejection decline under ultrahigh pressures. This performance enables single-stage concentration of brine beyond 180 g/L NaCl, reducing the need for complex multistage systems and associated capital costs[15]. At 200 bar and 50% recovery, the specific energy consumption (SEC) of the TFX membrane system is ~9.7 kWh/m³, assuming 90% pump efficiency and 95% energy recovery efficiency[15]. A recent study identified membrane water permeance as a top-value innovation target in high-salinity applications and showed that performance degradation—such as that from compaction—can drive sharp increases in LCOW unless mitigated by durable material designs[41]. Thus, the thermoset architecture of TFX membranes not only ensures mechanical integrity but also delivers measurable energy and cost savings in high-recovery brine concentration processes.

To enable large-scale production of TFX membranes, existing TFC membrane manufacturing lines can be strategically adapted with minimal modification. As illustrated in Fig. 3f, the conventional TFC production process—which typically involves casting a polymer solution onto a nonwoven fabric followed by coagulation in a nonsolvent bath—can be seamlessly integrated with in-line chemical crosslinking of the support layer. After the formation of the crosslinked support, the membrane passes through subsequent interfacial polymerization steps involving immersion in aqueous amine and organic acyl halide solutions, followed by thermal curing in an oven and post-treatment. This modular adjustment to the TFC roll-to-roll process enables the formation of a fully thermoset composite structure, characteristic of TFX membranes, without disrupting the overall production flow. By leveraging the existing industrial infrastructure, this approach facilitates rapid translation of TFX technology from lab-scale to scalable, industrial membrane fabrication.

### *In operando* characterization and quantitative evaluation of composite membranes before and after compaction

Cross-sectional SEM images of *e-TFX* (Fig. 4a, b) and *i-TFX* (Fig. 4c, d) membranes before and after compaction at 200 bar reveal minimal structural change, while the commercial TFC HPRO membrane (Fig. 4e, f) shows a significant loss of porosity and a marked reduction in thickness. Non-destructive, *in operando* SEM imaging was used to compare the high-pressure response of TFX and TFC membranes (up to 120 bar, limited by instrument capacity)[42]. The segmentation techniques quantified pore size distribution (PSD) and measured the thickness of the support layers in both TFX and TFC membranes. As illustrated in Supplementary Movies 1 and 2, the TFX membrane maintains structural integrity under pressures up to 120 bar, while the pore structure of the commercial TFC HPRO membrane deforms significantly with increased pressure.

According to the AI segmentation analysis (Supplementary Movies 3 and 4), the PSD data presented in Fig. 4g, h, and i indicate that the pore size distribution of the TFX membrane remains stable under varying pressure levels, exhibiting only a 2.9% compaction at pressures up to 120 bar. In contrast, the TFC membrane experiences substantial pore densification and thickness reduction, with compaction at 120 bar resulting in a 37% decrease in thickness. This extensive compaction in the TFC membrane corresponds to its performance degradation, marked by a sharp decline in water permeance and an increase in salt permeability.

At 200 bar, the TFX membrane exhibits a thickness reduction of approximately 13%, in contrast to the commercial TFC membrane, which loses nearly 42% of its thickness under high-pressure conditions (Fig. 4j). Such significant thickness reduction in commercial TFC membranes can lead to structural deformations in spiral-wound elements, as the feed spacers may become misaligned or distorted due to gaps forming between the membrane leaves. These open areas can exacerbate fouling and scaling, ultimately compromising the operational efficiency of the element[4].

## Discussion

By transitioning from a thermoset coating film over a thermoplastic support to a fully thermoset composite membrane structure, TFX membranes represent a paradigm shift in RO membrane performance with potential applications ranging from low-pressure water recycling to ultrahigh pressure brine concentration. While this study should be considered a successful proof of concept, the performance of TFX membranes should improve over time through exploring different polymer chemistries, crosslinking agents, and coating film chemistries. This study highlights the exceptional compaction resistance and separation performance of TFX membranes, but more importantly, the tunability of these properties. For example, the introduction of TEA as an acid-scavenging catalyst in the IP process illustrates the ability to improve TFX membrane water permeance without sacrificing salt rejection. Future work should further explore different coating film monomers and chemistries (e.g., amides, esters, ureas) to determine

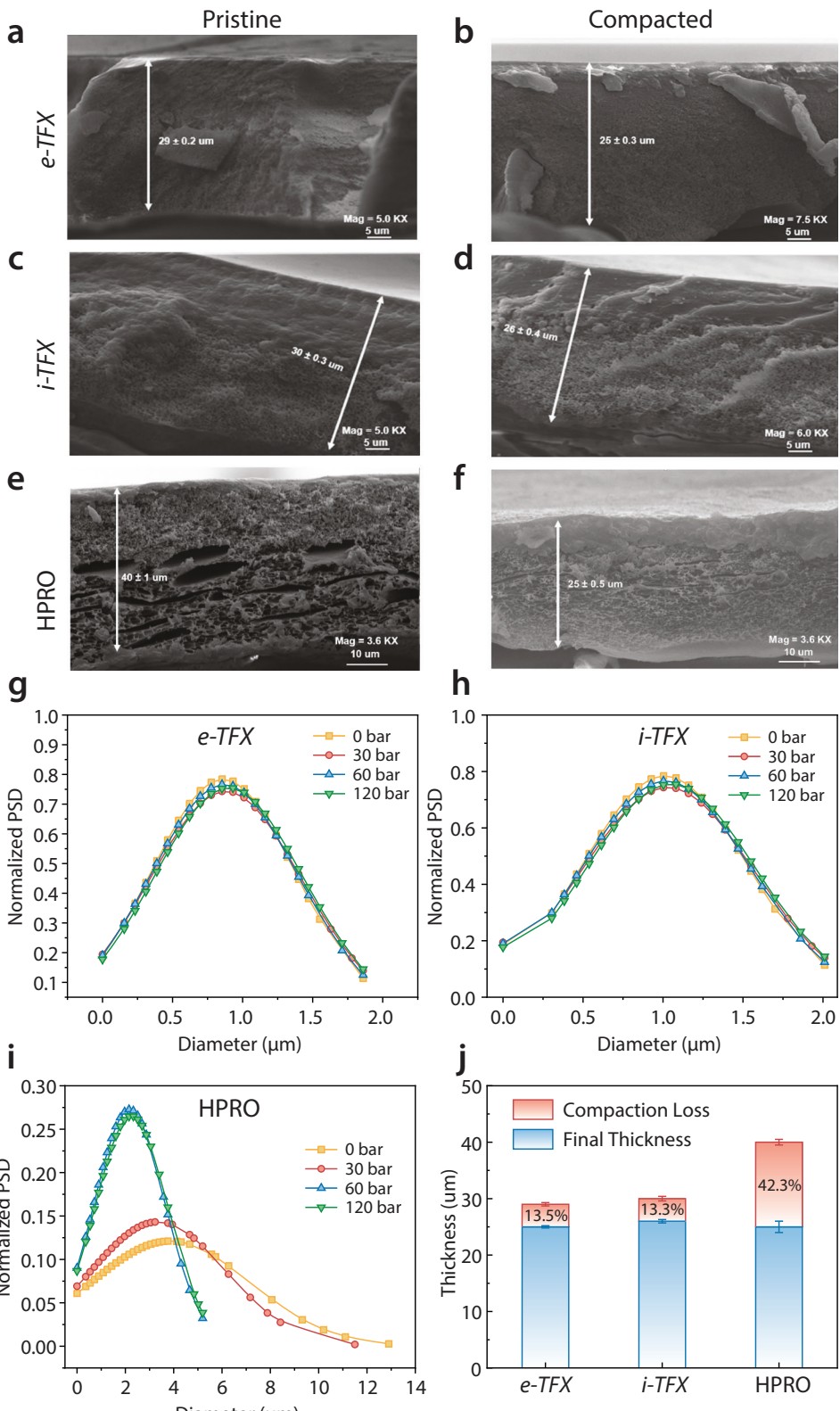

**Fig. 4 | Characterization and quantitative evaluation of composite membranes.** Cross-sectional SEM of *e-TFX*, **a** before and **b** after compaction at 200 bar. *i-TFX*, **c** before and **d** after compaction at 200 bar. Commercial HPRO **e** before and **f** after compaction at 200 bar (Reprinted from ref. 12, Copyright (2022), with permission from Elsevier). PSD of **g** *e-TFX*, **h** *i-TFX*, and **i** commercial HPRO at 120 bar. **j** Thickness of porous interlayers of TFX and TFC membranes before and after 200 bar compaction. Data represent the mean ± standard deviation from three independent measurements, where applicable.

the optimal combination of crosslinked support and coating film layers in TFX composite RO membrane structures.

Numerous crosslinking strategies for PI beyond the aliphatic 1,6-hexanediamine (HDA) could be explored. For example, more rigid aromatic diamines (e.g., p-phenylenediamine, xylylenediamine) and other chemistries should be explored to optimize membrane compaction resistance and performance. Alternative crosslinking reactions such as thermally- and UV-induced crosslinking could enable further fine-tuning of pore size, compaction resistance, and compatibility with different IP conditions. Also, the porous structure of the composite should be tailored for the intended applications. Fabrication parameters such as polymer concentration, solvent system, and coagulant composition modulate the support morphology, especially surface porosity and interfacial structure. Meanwhile, crosslinking chemically reinforces the bulk matrix, enhancing mechanical integrity. In future membrane designs, this interplay can be tailored to meet different performance needs. For example, in high-flux applications, a more porous interfacial layer may be beneficial for permeability, while maintaining a crosslinked core to preserve structural stability under high pressure.

Exploring alternative polymer backbone chemistries and crosslinking chemistries offers even more opportunities to enhance the performance of TFX membranes. While PI was used here as the support layer polymer, other polymer backbones crosslinked via appropriate chemical routes with more rigid crosslinking molecules should also be investigated to refine the properties and performance of TFX membranes. Additionally, these approaches provide greater degrees of freedom to design and manufacture TFX membranes, extending their potential range of applications.

## Methods

### Materials
Polyimide (PI) polymer (P84) was provided by Ensinger Sintimid GmbH (Austria). All solvents used were HPLC grade. Isopropanol (IPA), hexane, 1,4-dioxane, N,N-dimethylformamide (DMF), trimesoyl chloride (TMC) 98%, m-phenylenediamine (MPD) flakes > 99%, 1,6-hexanediamine (HDA), polyethylene glycol (PEG), triethylamine (TEA), and citric acid were purchased from Sigma Aldrich (St. Louis, Missouri, USA). MPD and TMC were used as monomers for the formation of the PA active layer via an interfacial polymerization process, using 18-MΩ ultrapure de-ionized water and hexane as aqueous and organic phases, respectively. The nonwoven fabric for membrane casting was provided by AZTECH Corporation.

### Ex situ crosslinked PI support membrane preparation
Polymer dope solutions were prepared by dissolving 16%–22% (wt%) PI in DMF (and with 1,4-dioxane as a co-solvent), and stirred overnight until complete dissolution was obtained. Solvent-to-co-solvent ratios of 1:1, 1:2, and 1:3 were examined. Once dissolved, the dope solution was placed in a fume hood for 10 h to remove entrapped air from the dope. The dope solution was then cast onto a polyester nonwoven fabric taped to a glass plate using a casting knife (Elcometer, US) set at a thickness of 150 μm. The knife was moved at a consistent casting speed of 0.1 m/s. After drawing out the dope solution was let sit for about 5 s before the film was immersed in the nonsolvent bath (either DI water or 100% IPA). After 30 min, the membrane was immersed in a fresh DI bath and left for 1 h to ensure sufficient removal of the solvents and stability of the membrane's final structure. Some of the support membranes were stored for PI-TFC membrane preparations, while others were further processed for ex situ crosslinking. The wet membrane was then immersed in a pure IPA bath for 1 h to displace water and solvents with IPA. The dewatered membrane was then placed in the 20 g/L HDA-IPA bath for crosslinking for 24 h. All crosslinked membranes were washed with IPA to remove excess crosslinker and stored in DI water before a PA coating was applied via interfacial polymerization.

### In situ crosslinked PI support membranes preparation
After the dope solution with a desired polymer concentration (16%–20%) was prepared, and degassed. The solution was then cast onto a polyester nonwoven fabric exactly as described above. Again, waiting 5 s after drawing out the dope solution over the fabric, the film was then immersed in a 20 g/L HDA in 100% IPA bath for simultaneous phase inversion and crosslinking. The precipitated membrane was left to sit in the bath for 30 min. The in situ crosslinked support membranes were immersed in IPA for 1 h to remove any unreacted HDA molecules, and then immersed and washed with DI water three times and stored in DI water prior to the interfacial polymerization process.

### Interfacial polymerization process
A support membrane (15 cm × 15 cm) taped to a glass plate (20 cm × 20 cm) was clamped under a customized polypropylene frame (10 cm × 10 cm × 3 cm). Then 250 mL aqueous solution of 6.0% (w/w) MPD was added to the frame where the support membrane sat underneath. For t-TFX preparation, ex situ TFX 2 (Supplementary Table 2) was used as the support, and 2.0% TEA (w/w) was added to the MPD solution. After 2 min, the MPD solution was removed along with the frame. The support membrane was air-knifed (Exair, Cincinnati, OH) with purified air to remove MPD solution, leaving the support membrane surface dry. And then a clean polypropylene frame (10 cm × 10 cm × 3 cm) was clamped atop the MPD-coated support. A solution of 0.15% (w/w) TMC in hexane was added to the frame. After 1 min coating time, the solution and the frame were removed, and the coated membrane remained on the glass plate for about 1 min before curing. The coated membrane was then hung vertically on a metal frame in the center of the oven (737 F Isotemp Oven, Fisher Scientific) and let sit at 90 °C for 3 min (the oven was preheated to the cure temperature before the membrane was hung). Then the coated membrane was soaked in a 60 °C citric acid bath at pH 3 for 15 min. Last, the membrane was soaked in a 60 °C DI bath for 15 min. After the curing steps, the membrane was stored in DI water at room temperature prior to wet-testing and characterization.

### Membrane testing
A commercially available, TFC HPRO membrane (Dupont XUS1808, HPRO) was tested along with hand-cast PI-TFX, PI-TFC, and PSU-TFC membranes in this work. Flux and rejection measurements were made using a rapidly-stirred, dead-end filtration cell (HP4750X Hastelloy Stirred Cell, Sterlitech Corp). All filtration experiments were performed in triplicate under each testing condition. Reported values represent the average, and error bars indicate the standard deviation of the three measurements. The schematic of the experimental filtration apparatus was described in detail previously[12]. All membranes were tested at ultrahigh pressure, up to 200 bar, supplied by high-pressure $N_2$ gas (Airgas USA, Radnor, Pennsylvania, USA). Laboratory 18 MΩ de-ionized (DI) water was used in pure water permeability experiments; volumetric water flux was determined by collecting permeate in a plastic cup resting on a digital balance (OHAUS Pioneer® Precision, Ohaus Corp., Parsippany, New Jersey, USA). The filter cell accommodated 49 mm diameter membrane coupons and 300 mL of aqueous NaCl feed solution.

In salt rejection tests, different concentrations of NaCl (Sigma Aldrich, S7653) were prepared as feed solution to maintain ~30 bar trans-membrane hydraulic pressure. For example, at 200 bar, the osmotic pressure of the feed solution was set at 170 bar.

The osmotic pressure of the feed solution was calculated via the following polynomial fit to data produced by a commercial geochemical modeling software (OLI Systems, Inc., Parsippany, NJ, USA):

$$\pi = 0.741829 \cdot c + 0.00111004 \cdot c^2 \tag{1}$$

where $\pi$ is the osmotic pressure (bar), and $c$ denotes salt concentration in g-NaCl/L-H$_2$O.

Water flux $J_w$ and salt flux $J_s$ are determined from

$$J_w = A(\Delta p_m - \Delta \pi_m) \qquad (2)$$

$$J_s = B\Delta c_m \qquad (3)$$

Here, $A$ indicates the observed water permeance of the membrane, $\Delta p_m$ is the trans-membrane hydraulic pressure, $\Delta \pi_m$ is the osmotic pressure difference across the membrane, $B$ is the NaCl permeance of the membrane, and $\Delta c_m$ is the real concentration difference across the membrane.

The concentration polarization factor ($CP$) is calculated according to[43,44]

$$CP = \frac{c_m}{c_b} = 1 - R_{ob} + R_{ob} \exp\left(\frac{J_s}{k_s}\right) \qquad (4)$$

where $R_{ob}(=1 - \frac{c_p}{c_f})$ is the observed salt rejection, $k_s$ is the solute mass transfer coefficient, $c_m$ is the membrane surface salt concentration, $c_p$ is the permeate salt concentration, and $c_f$ is the (bulk) feed salt concentration. For the turbulent flow in a stirred cell (Re > 32,000), the mass transfer coefficient is related to the Sherwood number, $Sh$, according to

$$k_s = Sh\frac{D}{d} \qquad (5)$$

$$Sh = 0.04 \cdot \text{Re}^{0.75} \cdot Sc^{0.33} \qquad (6)$$

and

$$\text{Re} = \frac{d^2 \cdot \omega \cdot \rho}{\mu} \qquad (7)$$

Here, $D$ is the diffusion coefficient, $d$ is the diameter of the stir cell, Re is the Reynolds number, $\omega$ is the rotation speed revolutions per second, $\rho$ is the liquid mass density, $\mu$ is the dynamic viscosity and $Sc$ is the Schmidt number ($= \nu/D$) where $\nu$ is the kinetic viscosity ($= \mu/\rho$) The intrinsic rejection $r_s$ and salt permeance can be determined by

$$r_s = 1 - \frac{(1 - R_{ob})}{CP} \qquad (8)$$

$$B = \frac{J_s(1 - r_s)}{r_s} \qquad (9)$$

Also, since the permeance, $A$, value of an RO membrane changes with the concentration of salt in the feedwater to which it is exposed[45], the $A$ value in the presence of salt water can be determined from Eq. (2) where

$$\Delta \pi_m = CP \cdot \pi_{c_f} - \pi_{c_p} \qquad (10)$$

The solute rejection by uncoated support membranes was determined using 100 kDa polyethylene glycol (PEG) at 1 g/L and 20 kDa PEG at 1 g/L, respectively. Solute concentrations in feed and filtrate were analyzed with a TOC-LCPH (total organic carbon) analyzer (Shimadzu Scientific Instruments, USA). Each type of membrane was tested 3 times to ensure accuracy. The stoke radius of the PEG solutes

was calculated based on their molecular weights as follows[38]:

$$R_{PEG} = 16.73 \cdot 10^{-3} \cdot M^{0.557} \qquad (11)$$

where $R$ is the radius of the PEG (in nanometers), respectively. $M$ represents the molecular weight of the particles (in Dalton). The solute-to-pore size ratio is known to be relevant to the solute rejection $r_s$ using the mechanical sieving model[46]:

$$r_{PEG} = [\lambda \cdot (2 - \lambda)]^2 \qquad (12)$$

where $\lambda$ ($= R_{PEG}/R_{Pore}$) is the solute-to-pore size ratio, $R_{PEG}$ is the solute radius, and $R_{Pore}$ is pore radius. Thereby, $R_{Pore}$ of the uncoated support membrane can be determined according to its rejection, $r_{PEG}$, of 100 kDA PEG solutions[47].

## Composite membrane physical-chemical characterization

Membrane samples were characterized using scanning electron microscopy (SEM) (Zeiss Supra 40 VP, Carl Zeiss Microscopy, LLC, NY, USA). For cross-sectional SEM imaging, the nonwoven backing fabric was carefully exfoliated from both pristine and tested membranes to expose the full thickness of the support and active layers. Image analysis was performed using NIH ImageJ software, where binarization and thresholding were applied to extract pore structures and quantify porosity, following the procedure described by Wu et al.[48].

*In operando* SEM characterization and AI segmentation of the composite membranes under pressure were performed using a CrossBeam 340 SEM (Carl Zeiss Microscopy, LLC, NY, USA) equipped with an *in operando* compression stage[42]. The schematic of the *in operando* SEM imaging setup is included in Fig. S11. Membrane coupons with a clean cross-sectional area of at least 20,000 μm × 50 μm were carefully prepared and mounted within the compression chamber. Compression was applied perpendicular to the membrane surface, with pressure gradually increased to 30, 60, and 120 bar. High-resolution SEM images were recorded at each pressure level to visualize morphological changes under realistic operating conditions[49].

Atomic Force Microscopy (AFM) analysis was performed using Park NX20 AFM (Park Systems, Santa Clara, CA, USA) to assess the surface roughness of the TFX membrane samples. During the experiments, the membrane samples were dried in a vacuum overnight before being adhered to silicon wafers with carbon tape, followed by non-contact mode measurements with a setpoint height of 9.5 μm. Scanned areas of 5 × 5 μm$^2$ were performed for each sample, and the results were included in the SI. The average roughness (R$_a$), root-mean-square roughness (R$_q$), and surface skewness (S$_{sk}$) were calculated by the data analysis software XEI according to ISO 4288.

Contact angle measurements were conducted using a contact angle goniometer (Model: 250, ramé-hart instrument co., Succasunna, New Jersey, USA).

FTIR Spectroscopy (Agilent Cary 630, Agilent Technologies, Santa Clara, CA, USA) was used to confirm the crosslinking of PI with HDA. X-ray Photoelectron Spectrometer (XPS) (K-Alpha XPS, Thermo Fisher Scientific Inc., Waltham, MA, USA) was used to determine the atomic composition of the membrane active layer.

The crosslinking degree of the PA film was calculated based on the O/N elemental ratio from the XPS spectra. To calculate the crosslinking degree, the following equation was used[50]:

$$\frac{O}{N} = \frac{3m + 4n}{3m + 2n}, \text{ and } m + n = 1 \qquad (13)$$

where $m$ and $n$ are the crosslinked and linear portions, respectively. The crosslinking degree is then obtained by $m \times 100\%$.

Differential scanning calorimetry (DSC) (PerkinElmer Inc. DSC-8500) was used to determine the glass transition temperature (Tg) of different polymers. These analyses were carried out at temperatures of between 30 °C and 450 °C at a heating and cooling rate of 10 °C/min. The experiments were performed under a nitrogen atmosphere at a flow rate of 20 °C/min. The Tg values of the polymers were taken from the second heating scan.

Membrane samples prepared without nonwoven fabric were used for tensile tests using the Instron 5944 universal testing machine (Instron Corporation, Norwood, MA). The printed dog bone specimens were elongated between the stationary and moving clamps. The applied loads were measured using the Instron load cell, which had a load capacity of 2000 N, while the displacements were measured using the integrated encoder linked to the crosshead movement. The stress–strain curves were subsequently computed based on the obtained data.

## Data availability
All data needed to evaluate the conclusions in the paper are present in the paper and/or the Supplementary Information. Additional data are available from the corresponding author upon request.

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

## Acknowledgements

The authors are grateful for the financial support for this study provided by the UCLA Samueli Engineering School, the UCLA Department of Civil & Environmental Engineering, and the UCLA Sustainable LA Grand Challenge. This material is based upon work supported by the National Alliance for Water Innovation (NAWI), funded by the U.S. Department of Energy, Office of Energy Efficiency and Renewable Energy (EERE), Advanced Manufacturing Office, under Funding Opportunity Announcement Number DE-FOA-0001905.

## Author contributions

J.W. and E.H. conceptualized and designed the study. J.W. and J.Q. conducted experimental research. J.W., M.X., and F.C. performed imaging. J.W., Y.C., H.F., X.W., and J.Q. performed data analysis and visualization. S.S., S.N.A., H.S., M.D., Y.L., D.J., J.R.M., and M.E. contributed to data interpretation and discussion. All authors discussed the results. E.H. supervised the study. J.W. wrote the manuscript. All authors, including J.W., J.Q., J.H., M.X., Y.C., H.F., X.W., F.C., K.P., Y.S., S.S., S.N.A., H.S., M.D., Y.L., D.J., J.R.M., M.E., and E.H., contributed to manuscript editing.

## Competing interests

The authors declare no competing interests.
