## [Transparent Peer Review file · Nature Communications]

Ultrahigh Pressure Compaction-Resistant Thin Film Crosslinked Composite Reverse Osmosis Membranes

Corresponding Author: Professor Eric Hoek

Version 0:

Reviewer comments:

Reviewer #1

(Remarks to the Author)

The authors investigated the development of high pressure compaction resistant TFC membranes. They explored the ex situ and in situ crosslinking procedures to improve the membrane performances. They emphasized that forming a fully thermoset composite membrane structure (both support and active layer) via crosslinking is important to have a compaction resistance membrane. However, other parameters, including membrane morphology controlled by the fabrication parameters also matter. It is unclear how to decouple these effects and understand the interplay between these parameters. In my opinion, the manuscript lacks in depth discussion on important aspects of the research, the methodology needs to be further detailed, need consistent labeling. The manuscript is more suitable for a specialized journal focusing on membranes after addressing the following comments:

1. In the experimental part, please elaborate in detail how the in operando SEM under pressure measurement was performed, not only refer to the reported literature, including the dimension of the samples, etc.
2. It is not clear how the authors quantified the compaction resistance. Since this is the main focus of the paper, a clear methodology should be provided, either by measuring the cross-section SEM or by comparing the flux measurement from the filtration experiment.
3. Methodology to determine the crosslinking degree must be detailed.
4. In page 4 lines 145-14, you write that increasing casting solution makes the support membrane less hydrophilic. Explain in the manuscript why this happens as the chemistry remain the same regardless the casting solution concentration.
5. The authors should explain how both ex situ and in situ crosslinking reduced the support surface porosity, and how the surface porosity is quantified should be detailed.
6. In page 6 lines 191-193, explain why the in situ crosslinked TFC yielded superior membrane performance than ex situ crosslinked TFC.
7. In Fig. 1e and 1f, explain in the manuscript how the same co-solvent ratio considerably affects the water permeance of ex situ and in situ crosslinked support, but almost not affecting their rejection performances.
8. The rejection of PEG (neutral molecules) should be dominated by size sieving mechanism determined by the pore size of the membrane and not the membrane chemistry. The authors should explain in the manuscript why the decrease in rejection values (Fig. 1f) did not show a clear trend with an increase in the mean pore size (Fig. 1g).
9. In fig 2a and 2b, depending on the casting solution, co-solvent ratio, and crosslinking method (ex situ or in situ), the water permeance, rejection, and solute permeability varied and there is not clear trend. The authors should explain in more detail this data and draw a conclusion on the relation between each parameter to the observed results.
10. The authors should explain in the manuscript how different crosslinking procedure (ex situ and in situ) resulting different surface morphology (Fig. S7).
11. Thickness and roughness of the selective layer play important role in determining the separation performance. The authors should provide thickness and roughness information by using AFM technique.
12. In Figure 3c, the authors should explain why the compaction of t-TFX at higher pressure (120 and 200 bar) is worse than e-TFX and i-TFX.
13. t-TFX showed superior performance than the other studied TFX. However, the surface characterization (SEM, AFM, contact angle) of t-TFX is not provided. The authors should complement the characterizations to support the arguments.

14. It is not clear the interplay between membrane morphology (caused by different membrane fabrication parameters) and membrane chemistry (application of crosslinking) that influence the compaction resistance.
15. How is the compaction resistant of dense support compared to dense and porous support with ex situ and in situ crosslinking? This experiment is important to decouple the effect of membrane morphology and crosslinking.
16. It is not clearly stated if the tensile strength in Fig. 1b is the tensile strength of the PI cast on non-woven support or only free-standing PI. This will make a huge different, as the tensile strength will be more dominated by the non-woven support. The authors should measure the mechanical strength of the free-standing support membranes without non-woven fabric.
17. The authors should explain the rationale comparing in situ and ex situ crosslinking procedures from chemistry or molecular transport perspective.
18. In Fig. 1d, although the uncrosslinked PI exhibited more compaction than the crosslinked PIs, the flux of the uncrosslinked PI is still higher than the crosslinked PI. Moreover, both the uncrosslinked and crosslinked PI demonstrated continuous decline in flux for 2 hours test. This means the uncrosslinked PI is still more beneficial than crosslinked PI in terms of flux. Please comment on this matter and justify why crosslinked PI is needed.
19. In TFX membranes, can the authors identify if the compaction occurred to both thin layer and PI porous support?
20. The compaction of ex situ and in situ crosslinked support is very similar. So what is the advantage of different crosslinking procedure?
21. The membrane labeling must be consistent for clarity and readability. For example, in page 8 line 219-229, the authors referred to supplementary table 4 and mentioned TFC-18% and TFC-20%, but in supplementary table 4 these membrane labels are not available.
22. The authors should provide justification why 1,6-hexanediamine (HDA) was used as crosslinking agent over other possible crosslinker.
23. The authors should mention in the experimental section how the error bar in the filtration experiments was obtained.
24. All abbreviations, such as HPRO, must be expanded at the first appearance.

Reviewer #2

(Remarks to the Author)

The manuscript introduces a new class of fully crosslinked composite RO membranes, referred to as TFX membranes. These membranes are engineered to resist physical compaction under ultrahigh pressures up to 200 bar—a significant limitation in conventional RO technologies. The key innovation lies in replacing the conventional thermoplastic polysulfone (PSU) support layer with a crosslinked polyimide (PI) support, creating a fully thermoset membrane structure. This design significantly enhances mechanical strength and mitigates irreversible compaction, which is a major source of performance degradation in standard thin-film composite (TFC) membranes.

The authors present a well-structured investigation into membrane fabrication, optimizing key parameters such as polymer concentration, solvent composition, and crosslinking method. They compare ex situ and in situ crosslinking approaches for the PI support, both of which yield robust structures suitable for polyamide layer formation via interfacial polymerization. Performance is further enhanced by incorporating triethylamine (TEA) in the polymerization step, producing tailored TFX membranes with significantly improved water permeance, high salt rejection (>99%), and strong compaction resistance under extreme salinity and pressure. The study is supported by comprehensive characterization, including FTIR, XPS, SEM, and in operando imaging, which confirms the structural integrity of TFX membranes compared to commercial HPRO counterparts.

The study appears to be solid and provides important insights to developing high-pressure compatible membranes. There are a few points that the authors may want to consider:

1. L74-75, this claim on the universal benefit of compaction reduction across all applications has not been rigorously proven. We don't know exactly how much performance enhancement the proposed approach would bring to the low-pressure applications. Also, it has been argued that higher permeability membranes can only bring marginal improvement to the energy consumption and cost for SWRO.
2. Fig. 3a, it appears that HPRO membrane is still superior even at 120 bar despite the compaction. The advantage of t-TFX is evident at 200bar. This point should be acknowledged. Also, the authors attribute the low permeability of TFX membranes (vs. HPRO or PSU-TFC) to the dense support layer. It'd be helpful to provide a breakdown of resistance between the active and support layer for the different membranes under different pressures.
3. following the last two point, the authors should consider providing a high-level quantitative analysis regarding how the proposed membrane with reduced compaction can benefit high TDS brine concentration. E.g., <https://doi.org/10.1073/pnas.2022196118>
4. Fig. 3. The authors should consider reporting water flux, which goes with the observed rejection, as both are raw data, then A and B. In its current form it's not clear to the review that if the flux for HPRO and PSU-TFC goes up or down when pressure is increased. If the flux goes up for HPRO, with the near constant B value, rejection should go up as well. The authors should have also reported the concentration and osmotic pressure of the feed solution in the caption. Lastly, ob in Rob should be subscript.
5. the method section, especially the equations, is written with little care and full of minor errors. E.g., L444, Jsw should be J_s, same for L452, r_s and R_o are not consistent in capitalization, R_o here is Rob in the figure, eqn 10 is missing a preceding factor that should include an osmotic coefficient evaluated using the Pfitzer model for high salinity. R_PEG and r_PEG are not consistent in eq.11 and 12. The use of dot and cross multiplication signs is very random. Font setting is very careless, inconsistent use of font size and type, the authors should learn from papers in math and physics.

Reviewer #3

(Remarks to the Author)

In this original research work, the authors present an approach for the preparation of a new class of thin-film composite membranes with application in high-pressure reverse osmosis (HPRO). Membrane compaction is a well-known issue in RO technology and is currently regarded as one of the major limitations for achieving higher recovery in desalination and brine concentration. As described by the authors, membrane compaction is tightly connected to the thermoplastic nature of the support membrane (typically PSU-PET) and the presented approach relies on the transition from a thermoset PA coating film over a thermoplastic PSU-PET support membrane to a fully thermoset composite membrane (TFX).

Overall, the manuscript is very well written and organized with a clear line of argument. The presented TFX membranes exhibit high compaction resistance and maintain a salt retention of over 99 % for NaCl up to 180 g/L as well as a satisfying water permeance at a pressure of 200 bar. Advanced imaging techniques show that, in contrast to commercial TFC-HPRO membranes, the TFX membranes deform significantly less during and after compression up to 200 bar. This finding is further supported by the measurement of pore size distributions before and after compaction. It is worth mentioning that the presented approach can be scaled up by a moderate upgrade of the current reverse osmosis membrane production technology. With the help of systematic optimization, the authors provide a toolbox for fine-tuning both the uncoated and the coated membrane. The in-depth analysis of the membrane performance suggests an interesting separation behavior of the TFX during compaction. The rejection of some membrane samples was improved during compaction.

The key point of this work relies in the interesting and very convincing approach to compaction resistant RO membranes that should be optimized for ultra-high pressure operations. Conventional development approaches aim to maintain membrane performance over a wide pressure range. Here the authors suggest to focus the optimization efforts on the particular conditions of the intended application namely ultra-high pressure. It is not about maintaining the performance but playing with the membrane dynamics to reach optimal performance at the desired pressure. The methodology is sound and presented in detail so that the experimental work can be reproduced. I could not find any flaws in the data analysis, interpretation and conclusions.

The excellent compaction resistance of the resulting TFX membranes up to 200 bar (<10% decline in water permeance) is particularly noteworthy. With a maximal operating pressure of 120 bar, Dupont XUS 1808 polyamide composite reverse RO membrane modules currently set the benchmark in HPRO operations. From this perspective, a shift in maximal operating pressure from 120 bar to 200 bar represents a very significant step in this field. The impact of this research goes beyond desalination and brine concentration. Indeed, such a progress in compaction resistance opens up new perspectives for HPRO, particularly in the areas of resource recovery & reuse from industrial wastewater as well as zero liquid discharge (ZLD). High-pressure batch RO processes that enable the simultaneous recovery/reuse of resources and production of clean water are well suited to closing loops, limiting liquid waste and preserving fresh water. The reutilization of recovered resources often requires a considerable concentration of brines, which leads to high osmotic and operating pressures. Such a batch RO process was recently successfully developed for the recovery and reuse of Cr(III) from electroplating wastewater with a specific energy consumption (SEC) of approx. 2.5 kWh/m³ (10.1016/j.desal.2024.117479 and 10.1021/acsestwater.4c00556). It is operated at pressures up to 120 bar and involves Dupont XUS 1808 RO membrane modules. In this case, process efficiency is clearly limited by the compaction resistance of the membrane and substantial improvement could be achieved by operating the system at pressure > 120 bar.

The authors presents an extensive dataset that reveals convincing evidence and strongly supports their claims and conclusions. The dataset combines independent characterization techniques ranging from advanced imaging techniques to the full determination of the properties and the performance (dry and wet-testing) of uncoated and coated membranes up to in operando characterization and quantitative evaluation of the composite membranes before and after compaction. The commercial Dupont TFC-HPRO membrane is the reference of choice for this work and the A200/A60 compaction resistance indicator is very well appropriate to demonstrate the gain in mechanical stability during the optimization the composite membranes for ultra-high pressure applications.

Overall, this is an impressive work and there is, from my side, very little to criticize.

1. I think there is a typo regarding the reference Dupont TFC-HPRO membrane that is referred as Dupont XUS 1818. I am not aware of this membrane type. To the best of my knowledge, the HPRO membranes is referred to as Dupont XUS 1808.
2. It would be helpful to provide a picture showing the fabrication flow of the most promising membrane and highlighting that the current state of the art can be adapted to host the new technology.
3. A few typos:
Line 77: the article the is repeated
Line 81: as an example of polymer or as an exemplary polymer
Line 98: systematic optimization.... This sentence has no verb.

Version 1:

Reviewer comments:

Reviewer #1

(Remarks to the Author)

The manuscript has been significantly revised to improve clarity by addressing most of the comments. However, there are still some minor concerns to address.

1. The authors have added the detail experimental procedure for the in operando SEM. However, a schematic or photo of the in operando SEM setup (instrument) under high pressure should be included either in the main manuscript or in the SI, as this nonconventional technique is only referenced via a PhD thesis that cannot be found online through the university repository search. A clear image would aid reader understanding, particularly for non-specialists.

2. Comment 7 remains partially unaddressed. The difference in permeance between in situ and ex situ treatments is more pronounced than the difference in rejection. For instance, 1:2-ex situ and 1:2-in situ shows permeance values of 20.6 and 16.4 LMH/bar, respectively (a 26% decrease), while their rejection rates remain nearly identical at 96.7% and 97.2%. It is worth considering that this likely stems from differences in surface and bulk effects: in situ treatment may influence both, leading to uniform densification, whereas ex situ primarily affects the surface, maintaining rejection performance but altering permeance.

3. The response to comment 10 needs more detail. The authors suggest that differences in crosslinking procedures (ex situ vs. in situ) affect surface morphology due to variations in monomer uptake and polyamide layer formation. However, they do not explain how these processes (monomer uptake and polyamide layer formation) occur in each method.

4. In the author contribution section, the authors did not mention the contribution of all authors. Moreover, in this revised version, a new author is added. Justify the need for adding new authors and detail the contribution of each author.

Reviewer #2

(Remarks to the Author)

the authors have satisfactorily addressed all my concerns. the manuscript is ready for publication.

Reviewer #3

(Remarks to the Author)

Thank you very much for the thorough revision of the manuscript, which can be published in the present form. Well done!

Version 2:

Reviewer comments:

Reviewer #1

(Remarks to the Author)

The remaining comments were also addressed, I recommend publishing the manuscript.

Reviewer 1

Comment 0:

The authors investigated the development of high pressure compaction resistant TFC membranes. They explored the *ex situ* and *in situ* crosslinking procedures to improve the membrane performances. They emphasized that forming a fully thermoset composite membrane structure (both support and active layer) via crosslinking is important to have a compaction resistance membrane. However, other parameters, including membrane morphology controlled by the fabrication parameters also matter. It is unclear how to decouple these effects and understand the interplay between these parameters. In my opinion, the manuscript lacks in depth discussion on important aspects of the research, the methodology needs to be further detailed, need consistent labeling. The manuscript is more suitable for a specialized journal focusing on membranes after addressing the following comments.

Our Response:

We sincerely thank the reviewer for the thoughtful and detailed assessment of our manuscript. We fully acknowledge the importance of decoupling the effects of membrane morphology and crosslinking chemistry, and we have now expanded our discussion to clarify how these factors individually and collectively influence compaction resistance and membrane performance. Specifically, we have added further analysis to distinguish the role of fabrication parameters (e.g., casting concentration, co-solvent ratio) from that of crosslinking procedures (*ex situ* vs. *in situ*) in shaping the support structure and surface morphology. Methodological details—such as sample dimensions for *in operando* imaging, crosslinking degree, and porosity quantification—have been added or clarified to improve reproducibility and clarity. In addition, we have revised the labeling of membrane samples throughout the main text and supplementary materials to ensure consistency.

We believe that, with these revisions, the manuscript now meets the rigor and interdisciplinary relevance suitable for the readership of *Nature Communications*.

Comment 1:

In the experimental part, please elaborate in detail how the *in operando* SEM under pressure measurement was performed, not only refer to the reported literature, including the dimension of the samples, etc.

Our Response:

Thanks for the comment. We agree that a clear description of the *in operando* SEM methodology is important. Detailed experimental procedures are included in the revised manuscript as follows:

Line 554 – 568: Membrane samples were characterized using scanning electron microscopy (SEM) (Zeiss Supra 40 VP, Carl Zeiss Microscopy, LLC, NY). For cross-sectional SEM imaging, the nonwoven backing fabric was carefully exfoliated from both pristine and tested membranes to expose the full thickness of the support and active layers. Image analysis was performed using NIH ImageJ software, where binarization and

thresholding were applied to extract pore structures and quantify porosity, following the procedure described by Wu et al¹.

In operando SEM characterization and AI segmentation of the composite membranes under pressure were performed using a CrossBeam 340 SEM (Carl Zeiss Microscopy, LLC, NY) equipped with an *in operando* compression stage². Membrane coupons with a clean cross-sectional area of at least 20,000 $\mu\text{m} \times 50 \mu\text{m}$ were carefully prepared and mounted within the compression chamber. Compression was applied perpendicular to the membrane surface, with pressure gradually increased to 30, 60, and 120 bar. High-resolution SEM images were recorded at each pressure level to visualize morphological changes at these compression pressures.

[1] Wu, J., Xiao, M., Quezada-Renteria, J. A., Hou, Z. & Hoek, E. M. Sample preparation matters: scanning electron microscopic characterization of polymeric membranes. *Journal of Membrane Science Letters*, 100073 (2024).

[2] Suleiman, Y. Characterization of Reverse Osmosis Membranes Utilizing In-Situ Multi-Scale Microscopy Ph.D. thesis, University of Connecticut, (2024).

Comment 2:

It is not clear how the authors quantified the compaction resistance. Since this is the main focus of the paper, a clear methodology should be provided, either by measuring the cross-section SEM or by comparing the flux measurement from the filtration experiment.

Our Response:

Thanks for your suggestion. In our study, compaction resistance is defined as the membrane's overall ability to retain performance at ultrahigh applied pressure, encompassing both **minimal loss in water permeance** and **stable salt rejection**. This performance stability is closely linked to the membrane's **structural integrity**, which we visualized via cross-sectional SEM before and after pressure testing (Fig. 4). Quantitatively, we use the ratio of water permeance at 200 bar versus 60 bar (A_{200}/A_{60}) as a primary metric (Fig. 2c). This definition captures the combined effects of mechanical deformation and transport stability, providing a robust indicator of pressure tolerance in high-salinity applications. Compaction resistance has now been defined in the revised manuscript:

Line 95 – 99: *In this study, we introduce a new approach to produce ultrahigh pressure tolerant RO membranes with excellent separation performance and compaction resistance—defined as the membrane's ability to retain water permeance and salt rejection under elevated hydraulic pressure, reflecting its structural integrity and performance stability during high-pressure operation.*

Comment 3:

Methodology to determine the crosslinking degree must be detailed.

Our Response:

Thank you for the comment. We have now clarified the methodology used to determine the crosslinking degree in the revised manuscript:

Line 536 – 541: The crosslinking degree of the PA film was calculated based on the O/N elemental ratio from the XPS spectra. To calculate the crosslinking degree, the following equations was used¹:

$$\frac{O}{N} = \frac{3m+4n}{3m+2n}, \text{ and } m + n = 1 \quad (13)$$

where m and n are the crosslinked and linear portions, respectively. The crosslinking degree is then obtained by $m \times 100\%$.

[1] Kim, S. H., Kwak, S.-Y. & Suzuki, T. Positron annihilation spectroscopic evidence to demonstrate the flux-enhancement mechanism in morphology-controlled thin-film-composite (TFC) membrane. *Environ. Sci. Technol.* **39**, 1764-1770 (2005).

Comment 4:

In page 4 lines 145-14, you write that increasing casting solution makes the support membrane less hydrophilic. Explain in the manuscript why this happens as the chemistry remain the same regardless the casting solution concentration.

Our Response:

Great point. As seen in Fig. S1 and Supplementary Table 1, increasing the polymer concentration in the casting solution results in a denser support structure with lower surface porosity. Although the membrane chemistry remains unchanged, this reduction in porosity leads to increased hydrophilicity, as fewer air-filled voids reduce water contact angles. This observation aligns with prior findings by Xiao et al.¹, who demonstrated that surface porosity strongly influences wettability: higher porosity membranes tend to exhibit more hydrophobic behavior, whereas reduced porosity enhances apparent hydrophilicity. We have revised the manuscript to clarify this:

Line 151 – 154: As the polymer concentration in the casting solution increases, the resulting support membrane becomes denser due to the decreasing surface porosity, which is attributed to slower phase inversion kinetics. This reduction in porosity leads to increased hydrophilicity, as fewer air-filled voids reduce water contact angles¹.

[1] Xiao, M., Yang, F., Im, S., Dlamini, D.S., Jassby, D., Mahendra, S., Honda, R. and Hoek, E.M., 2022. Characterizing surface porosity of porous membranes via contact angle measurements. *Journal of Membrane Science Letters*, 2(1), p.100022.

Comment 5:

The authors should explain how both *ex situ* and *in situ* crosslinking reduced the support surface porosity, and how the surface porosity is quantified should be detailed.

Our Response:

Thank you for this important comment. Both *ex situ* and *in situ* crosslinking reduce surface porosity by forming covalent bonds between polymer chains, which restricts chain mobility and densifies the polymer network. This densification occurs predominantly near the membrane surface due to limited crosslinker diffusion in the *in situ* case and surface-localized reactivity in the *ex situ* case. As illustrated in Supplementary Fig. S3, crosslinked

PI supports show visibly reduced surface porosity compared to their uncrosslinked counterparts, particularly under high polymer concentrations and slow phase inversion conditions.

This observation is well supported by literature. Vanherck et al. (2013) reviewed the effect of crosslinking on polyimide membranes and concluded that both diamine crosslinking and thermal treatment lead to reduced free volume, lower surface porosity, and densification of the membrane structure, with a corresponding decline in permeability due to tighter packing and reduced pore interconnectivity¹. In our study, the same densification behavior is observed as both crosslinking strategies limit macrovoid formation and surface pore development.

We have clarified these points in the revised manuscript:

Line 161 – 166: Both *ex situ* and *in situ* crosslinking reduced surface porosity by promoting polymer network densification near the surface, as shown in Supplementary Fig. S3. This effect is consistent with prior findings that crosslinking restricts chain mobility and reduces free volume, yielding tighter and less porous structures¹. Denser interfacial layers were also observed at high polymer concentrations, coagulation using IPA instead of DI, and high co-solvent ratios (Supplementary Fig. 5i and 5j and 6d to 6f).

Porosity measurements have been clarified in the Methods section:

Line 557 – 560: Image analysis was performed using NIH ImageJ software, where binarization and thresholding were applied to extract pore structures and quantify porosity, following the procedure described by Wu et al².

[1] Vanherck, K., Koeckelberghs, G., & Vankelecom, I. F. J. (2013). *Crosslinking polyimides for membrane applications: A review*. *Prog. Polym. Sci.*, **38**, 874–896. <https://doi.org/10.1016/j.progpolymsci.2012.11.001>

[2] Wu, J., Xiao, M., Quezada-Renteria, J. A., Hou, Z. & Hoek, E. M. Sample preparation matters: scanning electron microscopic characterization of polymeric membranes. *Journal of Membrane Science Letters*, 100073 (2024).

Comment 6:

In page 6 lines 191-193, explain why the *in situ* crosslinked TFC yielded superior membrane performance than *ex situ* crosslinked TFC.

Our Response:

While the *in situ* process improves fabrication efficiency, the superior desalination performance of *i-TFX* compared to *e-TFX* is primarily due to differences in the resulting active layer, rather than the support itself. Active layer formation is highly sensitive to support properties such as surface porosity, pore size, and roughness, which collectively influence amine monomer uptake and the effective amine-to-acyl chloride ratio during interfacial polymerization. As shown in Fig. S7, the surface morphologies of the polyamide

layers differ among TFX membranes, despite all being coated with the same formulation—this confirms that subtle variations in support structure directly impact the polyamide layer.

We have added the following discussion to the manuscript:

Line 212 – 218: The superior performance of *i-TFX* is attributed to the favorable match between its support structure and the chosen interfacial polymerization conditions. Although all TFX membranes were coated using the same formulation, the resulting active layer morphologies differed (Fig. S7), highlighting how subtle differences in support properties can affect monomer uptake and polyamide layer development. This underscores the importance of optimizing support–coating compatibility for achieving peak membrane performance.

[1] Dlamini, D. S. *et al.* On the role of the porous support membrane in seawater reverse osmosis membrane synthesis, properties and performance. *Journal of Membrane Science* **708**, 123032 (2024).

Comment 7:

In Fig. 1e and 1f, explain in the manuscript how the same co-solvent ratio considerably affects the water permeance of *ex situ* and *in situ* crosslinked support, but almost not affecting their rejection performances.

Our Response:

Thank you for the comment. We would like to clarify that Fig. 1e shows a consistent trend: both *in situ* and *ex situ* crosslinking reduce water permeance due to polymer densification, while improving rejection due to smaller effective pore sizes. The observed differences between *in situ* and *ex situ* crosslinked supports at the same co-solvent ratio likely arise from how the crosslinking step interacts with the evolving membrane structure. In *in situ* crosslinking, phase inversion and crosslinking occur simultaneously, allowing the crosslinker to access the polymer network while it is still semi-coagulated, potentially leading to a denser surface. In contrast, *ex situ* crosslinking occurs after the membrane structure is fixed, which may lead to less uniform densification.

Comment 8:

The rejection of PEG (neutral molecules) should be dominated by size sieving mechanism determined by the pore size of the membrane and not the membrane chemistry. The authors should explain in the manuscript why the decrease in rejection values (Fig. 1f) did not show a clear trend with an increase in the mean pore size (Fig. 1g).

Our Response:

Thank you for the comment. We would like to clarify that there is in fact a clear trend of increasing PEG rejection with decreasing mean pore size, as shown in Fig. 1f and 1g. This observation is consistent with size sieving principle:

Line 526 – 539: The solute rejection by uncoated support membranes was determined using 100 kDa PEG at 1 g/L and 20 kDa PEG at 1 g/L, respectively. Solute concentrations in feed and filtrate were analyzed with TOC-LCPH (total organic carbon) analyzer (Shimadzu Scientific Instruments, USA). Each type of membrane was tested 3 times to ensure accuracy. The stoke radius of the PEG solutes were calculated based on their molecular weights as follows¹:

$$R_{PEG} = 16.73 \times 10^{-3} \times M^{0.557} \quad (1)$$

where R is the radius of the PEG (in nanometer), respectively. M represents the molecular weight of the particles (in Dalton). The solute-to-pore size ratio is known relevant to the solute rejection r_s using the mechanical sieving model²:

$$r_{PEG} = [\lambda(2 - \lambda)]^2 \quad (2)$$

where $\lambda = (R_{PEG}/R_{pore})$ is the solute-to-pore size ratio, R_{PEG} is solute radius, and R_{pore} is pore radius. Thereby R_{pore} of the uncoated support membrane can be determined according to its rejection, r_{PEG} , of 100 kDa PEG solutions³.

[1] Ghosh, A. K. & Hoek, E. M. Impacts of support membrane structure and chemistry on polyamide–polysulfone interfacial composite membranes. *Journal of membrane science* **336**, 140-148 (2009).

[2] Mueller, J. & Davis, R. H. Protein fouling of surface-modified polymeric microfiltration membranes. *Journal of membrane Science* **116**, 47-60 (1996).

[3] Minhao Xiao, X. W., Ziwei Hou, Javier Alan Quezada Renteria, Derrick S. Dlamini, David Jassby, Eric M. V. Hoek. *Comparison of classical hydrodynamic pore-flow models of transport through porous membranes* (Separation and Purification Technology, 2024).

Comment 9:

In fig 2a and 2b, depending on the casting solution, co-solvent ratio, and crosslinking method (ex situ or in situ), the water permeance, rejection, and solute permeability varied and there is not clear trend. The authors should explain in more detail this data and draw a conclusion on the relation between each parameter to the observed results.

Our Response:

Thank you for the comment. We agree that the performance variation observed in Fig. 2a and 2b reflects the complex interplay between casting conditions, co-solvent ratio, and crosslinking method. Although all membranes were coated using the same interfacial polymerization formulation, the underlying support membranes differed in porosity, surface morphology, and chemical structure—resulting from variations in fabrication and crosslinking. These differences affected amine monomer uptake and polyamide layer formation, leading to the observed variability in water permeance, rejection, and solute permeability. We have clarified this explanation in the revised manuscript to emphasize the critical role of support–coating compatibility in determining final membrane performance:

Line 214 – 218: Although all TFX membranes were coated using the same formulation, the resulting active layer morphologies differed (Fig. S7), highlighting how subtle differences in support properties can affect monomer uptake and polyamide layer development. This underscores the importance of optimizing support–coating compatibility for achieving peak membrane performance.

Line 420 – 426: Fabrication parameters such as polymer concentration, solvent system, and coagulant composition modulate the support morphology, especially surface porosity and interfacial structure. Meanwhile, crosslinking chemically reinforces the bulk matrix, enhancing mechanical integrity. In future membrane designs, this interplay can be tailored to meet different performance needs. For example, in high-flux applications, a more porous interfacial layer may be beneficial for permeability, while maintaining a crosslinked core to preserve structural stability under high pressure.

Comment 10:

The authors should explain in the manuscript how different crosslinking procedure (ex situ and in situ) resulting different surface morphology (Fig. S7).

Our Response:

Thank you for the suggestion. While all TFX membranes were coated using the same interfacial polymerization conditions, the resulting surface morphologies (Fig. S7) varied depending on the underlying support structure¹, which was influenced by the crosslinking method. We have now clarified this point in the revised manuscript:

Line 214 – 218: Although all TFX membranes were coated using the same formulation, the resulting active layer morphologies differed (Fig. S7), highlighting how subtle differences in support properties can affect monomer uptake and polyamide layer development. This underscores the importance of optimizing support–coating compatibility for achieving peak membrane performance.

[1] Dlamini, D. S. *et al.* On the role of the porous support membrane in seawater reverse osmosis membrane synthesis, properties and performance. *Journal of Membrane Science* **708**, 123032 (2024).

Comment 11:

Thickness and roughness of the selective layer play important role in determining the separation performance. The authors should provide thickness and roughness information by using AFM technique.

Our Response:

Agreed. We have now included AFM measurements in our study to provide quantitative information on the surface roughness and estimate the thickness of the selective layer. The results are presented in the revised Supplementary Information (Fig. S9), and discussed in the manuscript to support the observed separation performance:

Line 569 – 576: Atomic Force Microscopy (AFM) analysis was performed using Park NX20 AFM (Park Systems, Santa Clara, CA, USA) to assess the surface roughness of the TFX membrane samples. During the experiments, the membrane samples were dried in vacuum overnight before being adhered to silicon wafers with carbon tape, followed by non-contact mode measurements with setpoint height of 9.5 μm . Scanned areas of $5 \times 5 \mu\text{m}^2$ were performed for each samples, and the results are included in SI. The average roughness (R_a), root-mean-square roughness (R_q), and surface skewness (S_{sk}) were calculated by the data analysis software XEI according to ISO 4288.

Line 283 – 288: Supplementary Fig. S8 and Fig. S9 show the surface SEM and AFM images of the *t-TFX*, respectively. While SEM does not reveal distinct morphological differences between *t-TFX*, *e-TFX*, and *i-TFX*, AFM analysis shows that *t-TFX* has a higher surface roughness ($R_q = 114.7 \text{ nm}$) compared to *e-TFX* ($R_q = 81.2 \text{ nm}$) and *i-TFX* ($R_q = 64.6 \text{ nm}$). This suggests a thicker polyamide layer formed on the *t-TFX*, which may explain its enhanced separation performance.

Fig. S9 AFM images and roughness measurements of TFX membranes. (a) *i-TFX*; (b) *e-TFX*; (c) *t-TFX*.

Comment 12:

In Figure 3c, the authors should explain why the compaction of *t-TFX* at higher pressure (120 and 200 bar) is worse than *e-TFX* and *i-TFX*.

Our Response:

Thank you for the observation. As discussed in our recent study¹, both transmembrane pressure and water flux contribute to membrane compaction. The slightly greater compaction observed in *t-TFX* at higher pressures (120 and 200 bar) is likely due to its higher flux compared to *e-TFX* and *i-TFX*. The elevated water flux can generate additional mechanical stress within the selective and support layers, leading to more pronounced structural densification over time. We have clarified this point in the revised manuscript and cited the supporting literature accordingly:

Line 296 – 298: The slightly more compaction observed in *t*-TFX in contrast to *e*-TFX and *i*-TFX could be attributed to its higher flux (Supplementary Fig. S10)¹.

[1] Wu, J. et al. Role of Transmembrane Pressure and Water Flux in Reverse Osmosis Composite Membrane Compaction and Performance. *Environmental Science & Technology* 2025. DOI: [10.1021/acs.est.5c02618](https://doi.org/10.1021/acs.est.5c02618).

Comment 13:

t-TFX showed superior performance than the other studied TFX. However, the surface characterization (SEM, AFM, contact angle) of *t*-TFX is not provided. The authors should complement the characterizations to support the arguments.

Our Response:

Thank you for the suggestion. We have now incorporated the full surface characterizations and corresponding discussion of *t*-TFX into the manuscript. Contact angle data are included in Supplementary Table S6, SEM images in Fig. S8, and AFM images in Fig. S9.

Line 313 – 316: Contact angle measurements indicate that *t*-TFX exhibits greater hydrophilicity than *e*-TFX and *i*-TFX (Supplementary Table S6). In addition, Fig. 3d highlights a higher crosslinking degree in *t*-TFX, as verified by XPS analysis (also Supplementary Table S6), confirming the enhanced effectiveness of TEA in improving both the interfacial polymerization process and crosslinking density within the selective layer.

Line 283 – 288: Supplementary Fig. S8 and Fig. S9 show the surface SEM and AFM images of the *t*-TFX, respectively. While SEM does not reveal distinct morphological differences between *t*-TFX, *e*-TFX, and *i*-TFX, AFM analysis shows that *t*-TFX has a higher surface roughness ($R_q = 114.7$ nm) compared to *e*-TFX ($R_q = 81.2$ nm) and *i*-TFX ($R_q = 64.6$ nm). This suggests a thicker polyamide layer formed on the *t*-TFX, which may explain its enhanced separation performance.

Comment 14:

It is not clear the interplay between membrane morphology (caused by different membrane fabrication parameters) and membrane chemistry (application of crosslinking) that influence the compaction resistance.

Our Response:

The compaction resistance of TFX membranes results from a synergistic interplay between membrane morphology—tuned via fabrication parameters—and membrane chemistry, specifically the application of crosslinking to form a fully thermoset composite structure.

As shown in Fig. 1d and discussed in the manuscript, the crosslinked PI support exhibits minimal flux decline under pressure, confirming its dimensional stability. Figures 2 and 3 further highlight the superior performance of TFX membranes over conventional TFC counterparts, and *in operando* SEM imaging reveals negligible structural deformation in

the crosslinked support, validating its compaction resistance. We have now added new discussion:

Line 420 – 426: Fabrication parameters such as polymer concentration, solvent system, and coagulant composition modulate the support morphology, especially surface porosity and interfacial structure. Meanwhile, crosslinking chemically reinforces the bulk matrix, enhancing mechanical integrity. In future membrane designs, this interplay can be tailored to meet different performance needs. For example, in high-flux applications, a more porous interfacial layer may be beneficial for permeability, while maintaining a crosslinked core to preserve structural stability under high pressure.

Comment 15:

How is the compaction resistant of dense support compared to dense and porous support with ex situ and in situ crosslinking? This experiment is important to decouple the effect of membrane morphology and crosslinking.

Our Response:

Fig. 1g shows that the crosslinked membranes have slightly smaller mean pore sizes compared to their uncrosslinked counterparts. As shown in Fig. 1d, despite comparable initial permeance, the crosslinked supports exhibited significantly reduced flux decline over time. This demonstrates that the enhancement in compaction resistance primarily stems from the formation of covalent crosslinking networks within the support. While denser morphology may offer some short-term structural benefits, it is the chemical crosslinking that plays the dominant role in preserving membrane integrity under prolonged ultrahigh-pressure operation.

Comment 16:

It is not clearly stated if the tensile strength in Fig. 1b is the tensile strength of the PI cast on non-woven support or only free-standing PI. This will make a huge difference, as the tensile strength will be more dominated by the non-woven support. The authors should measure the mechanical strength of the free-standing support membranes without non-woven fabric.

Our Response:

Great point. And yes, we did the measurements with free-standing approach. Manuscript has been updated accordingly:

Line 594 – 600: Membrane samples prepared without non-woven fabric were used for tensile tests using the Instron 5944 universal testing machine (Instron Corporation, Norwood, MA). The printed dog bone specimens were elongated between the stationary and moving clamps. The applied loads were measured using the Instron load cell, which had a load capacity of 2000 N, while the displacements were measured using the integrated encoder linked to the crosshead movement. The stress–strain curves were subsequently computed based on the obtained data.

Comment 17:

The authors should explain the rationale comparing *in situ* and *ex situ* crosslinking procedures from chemistry or molecular transport perspective.

Our Response:

Thank you for the helpful suggestion. We have now expanded the explanation comparing *in situ* and *ex situ* crosslinking approaches from both chemical and molecular transport perspectives in the revised manuscript.

From a chemical standpoint, *ex situ* crosslinking involves immersing a pre-formed PI membrane into a diamine solution, where crosslinking predominantly occurs near the surface due to limited diamine diffusion into the dense polymer matrix. However, several studies have shown that this method still yields structurally robust membranes with sufficient crosslinking depth to resist compaction and swelling under pressure¹. In contrast, *in situ* crosslinking is carried out during phase inversion, where the diamine diffuses through the polymer network while it is still in a semi-coagulated state. This enables more homogeneous crosslinking across the membrane thickness and eliminates the need for a separate post-treatment step.

From a molecular transport perspective, both methods produce mechanically reinforced supports, but *in situ* crosslinking may introduce slightly tighter surface layers due to concurrent gelation and crosslinking at the membrane interface. Overall, both strategies improve dimensional stability, and the choice depends on processing flexibility and integration goals. This explanation has been added to the revised manuscript for clarity:

Line 166 – 173: From a structural standpoint, both *in situ* and *ex situ* crosslinking reduce surface porosity by promoting polymer densification, but their crosslinking profiles may differ due to diffusion kinetics. *Ex situ* crosslinking typically results in more surface-concentrated modification, while *in situ* crosslinking enables more uniform integration as the diamine diffuses through the semi-coagulated polymer during phase inversion. Despite similar porosity outcomes, the *in situ* method offers process advantages by combining phase inversion and crosslinking in a single step, reducing fabrication time and improving scalability.

[1] Vanherck, K. et al. A simplified diamine crosslinking method for PI nanofiltration membranes. *J. Membr. Sci.* **353**, 135–143 (2010). <https://doi.org/10.1016/j.memsci.2010.02.046>

Comment 18:

In Fig. 1d, although the uncrosslinked PI exhibited more compaction than the crosslinked PIs, the flux of the uncrosslinked PI is still higher than the crosslinked PI. Moreover, both the uncrosslinked and crosslinked PI demonstrated continuous decline in flux for 2 hours test. This means the uncrosslinked PI is still more beneficial than crosslinked PI in terms of flux. Please comment on this matter and justify why crosslinked PI is needed.

Our Response:

While it is true that the uncrosslinked PI initially exhibits higher water flux due to its looser structure and higher porosity, it also undergoes significant compaction, leading to a sharp decline in flux over time. In contrast, both *ex situ* and *in situ* crosslinked PI supports start with slightly lower flux but exhibit excellent structural stability, as evidenced by their minimal flux decline during the 2-hour test (Fig. 1d).

The purpose of crosslinking is not merely to maximize initial flux, but to ensure **long-term mechanical integrity and operational stability**, especially under high-pressure conditions. Crosslinking densifies the support structure and suppresses polymer chain mobility, thereby resisting compaction and maintaining membrane performance over extended operation. In practical applications—such as high-pressure brine concentration—flux stability is far more critical than absolute initial flux.

Comment 19:

In TFX membranes, can the authors identify if the compaction occurred to both thin layer and PI porous support?

Our Response:

Yes, both the polyamide active layer and the porous PI support layer undergo compaction in TFX membranes. Previous studies have shown that the active layer compacts under high pressure due to polymer chain rearrangement and relaxation phenomena¹, while the support layer contributes significantly to overall thickness reduction and loss in water permeability²⁻⁷.

In our current study, we observed that compaction in TFX membranes occurs in both layers but is significantly reduced compared to conventional TFC membranes. From performance testing, TFX membranes show minimal flux loss ($A_{200}/A_{60} \approx 0.9$), and cross-sectional SEM indicates only ~10% thickness reduction—both of which represent substantial improvements over typical commercial TFC membranes ($A_{200}/A_{60} \approx 0.3$; ~43% thickness reduction). This demonstrates the mechanical robustness of both the crosslinked PI support and the thermoset polyamide layer.

[1] Wu, J. et al. Polyamide Reverse Osmosis Membrane Compaction and Relaxation: Mechanisms and Implications for Desalination Performance. *J. Membr. Sci.* **122893** (2024). <https://doi.org/10.1016/j.memsci.2024.122893>

[2] Pendergast, M. T. M., Nygaard, J. M., Ghosh, A. K., & Hoek, E. M. V. Using nanocomposite materials to understand and control reverse osmosis membrane compaction. *Desalination* **261**, 255–263 (2010).

[3] Davenport, D. M. et al. Thin film composite membrane compaction in high-pressure reverse osmosis. *J. Membr. Sci.* **610**, 118268 (2020). <https://doi.org/10.1016/j.memsci.2020.118268>

[4] Wu, J. et al. Reverse osmosis membrane compaction and embossing at ultra-high pressure operation. *Desalination* **537**, 115875 (2022). <https://doi.org/10.1016/j.desal.2022.115875>

[5] Lim, Y. J., Nadzri, N., Lai, G. S., & Wang, R. Demystifying the compaction effects of TFC polyamide membranes in the desalination of hypersaline brine via high-pressure RO. *J. Membr. Sci.* **122950** (2024).

[6] Xu, C., Wang, Z., Hu, Y., & Chen, Y. Thin-film composite membrane compaction: Exploring the interplay among support compressive modulus, structural characteristics, and overall transport efficiency. *Environ. Sci. Technol.* (2024).

[7] Wu, J. et al. Role of transmembrane pressure and water flux in reverse osmosis composite membrane compaction and performance. *Environ. Sci. Technol.* (2025). <https://doi.org/10.1021/acs.est.5c02618>

Comment 20:

The compaction of *ex situ* and *in situ* crosslinked support is very similar. So what is the advantage of different crosslinking procedure?

Our Response:

While the resulting compaction resistance of the *ex situ* and *in situ* crosslinked membranes is similar, the *in situ* crosslinking method offers clear practical advantages in terms of process integration and fabrication efficiency. Specifically, *in situ* crosslinking enables simultaneous phase inversion and chemical crosslinking in a single step, eliminating the need for post-treatment steps and reducing total processing time and chemical consumption.

This is clarified in the revised manuscript as follows:

Line 116 – 121: Fig. 1a depicts the *in situ* and *ex situ* phase inversion and crosslinking processes. The *ex situ* crosslinking involves phase inversion followed by crosslinking, whereas in the *in situ* approach the PI was directly phase inverted in the crosslinking solution, so both phase inversion and crosslinking occur simultaneously. This simplifies the fabrication process and makes it more practical for scaling up to commercial manufacturing.

Comment 21:

The membrane labeling must be consistent for clarity and readability. For example, in page 8 line 219-229, the authors referred to supplementary table 4 and mentioned TFC-18% and TFC-20%, but in supplementary table 4 these membrane labels are not available.

Our Response:

Thanks for the suggestion. We have carefully reviewed and updated the membrane labeling in both the main text and supplementary materials to ensure consistency with Supplementary Table 4:

Line 244 – 254: Supplementary Table 4 compares non-crosslinked PI TFC membranes with crosslinked TFX variants, illustrating distinct differences in reverse osmosis performance under increasing pressures. TFC membranes with higher polymer concentrations—such as TFC 2 (18% PI) and TFC 3 (20% PI)—exhibit enhanced salt rejection at lower pressures but reduced water permeability compared to TFC 1 (16% PI), highlighting the typical trade-off between selectivity and permeability. TFC 4 (16% PI, IPA-processed) shows lower salt rejection, likely due to unfavorable interactions between the PI support and m-phenylenediamine during polyamide layer formation. TFC 11 (18% PI, 1:1 solvent-to-co-solvent ratio) retains high salt rejection at lower pressures but shows a significant decline at 200 bar, emphasizing the importance of crosslinking for maintaining structural and performance stability under ultrahigh-pressure operation.

Comment 22:

The authors should provide justification why 1,6-hexanediamine (HDA) was used as crosslinking agent over other possible crosslinker.

Our Response:

Thank you for the comment. This is a proof of concept study. HDA was selected due to its well-documented reactivity with polyimides and simple chemistry. HDA is a flexible aliphatic diamine that reacts with imide groups to form stable covalent amide linkages, promoting network densification and solvent resistance. This approach has been widely adopted for polyimide membranes in organic solvent nanofiltration (OSN) and composite membrane applications, where HDA-crosslinked membranes exhibit reduced swelling, enhanced mechanical strength, and improved long-term performance under aggressive solvent or pressure conditions¹⁻⁵.

That said, we agree that exploring alternative crosslinkers could offer further opportunities to tailor support properties and discussed the perspective in the manuscript:

Line 413 – 418: Numerous crosslinking strategies for PI beyond the aliphatic 1,6-hexanediamine (HDA) could be explored. For example, more rigid aromatic diamines (e.g., p-phenylenediamine, xylylenediamine) and other chemistries should be explored to optimize membrane compaction-resistance and performance. Alternative crosslinking reactions such as thermally- and UV-induced crosslinking could enable further fine-tuning of pore size, compaction resistance, and compatibility with different IP conditions.

[1] Soroko, I., Y. Bhole, A.G. Livingston. "Environmentally friendly route for the preparation of solvent resistant polyimide nanofiltration membranes." *Green Chemistry* 13.1 (2011): 162-168. <https://doi.org/10.1039/C0GC00155D>

[2] Gorgojo, P., M. F. Jimenez-Solomon, A. G. Livingston. "Polyamide thin film composite membranes on cross-linked polyimide supports: Improvement of RO performance via activating solvent." *Desalination* 344 (2014): 181-188. <https://doi.org/10.1016/j.desal.2014.02.009>

[3] Siddique, H., et al. "Pore preserving crosslinkers for polyimide OSN membranes." *Journal of Membrane Science* 465 (2014): 138-150. <https://doi.org/10.1016/j.memsci.2014.03.031>

[4] Vanherck, K. et al. A simplified diamine crosslinking method for PI nanofiltration membranes. *J. Membr. Sci.* 353, 135–143 (2010). <https://doi.org/10.1016/j.memsci.2010.02.046>

Comment 23:

The authors should mention in the experimental section how the error bar in the filtration experiments was obtained.

Our Response:

Thank you for the comment. We have clarified this point in the revised manuscript:

Line 493 – 495: All filtration experiments were performed in triplicate under each testing condition. Reported values represent the average, and error bars indicate the standard deviation of the three measurements.

Comment 24:

All abbreviations, such as HPRO, must be expanded at the first appearance.

Our Response:

Thank you for the suggestion. We have carefully revised the manuscript to ensure that all abbreviations are clearly defined at their first appearance.

Reviewer 2

Comment 0:

The manuscript introduces a new class of fully crosslinked composite RO membranes, referred to as TFX membranes. These membranes are engineered to resist physical compaction under ultrahigh pressures up to 200 bar—a significant limitation in conventional RO technologies. The key innovation lies in replacing the conventional thermoplastic polysulfone (PSU) support layer with a crosslinked polyimide (PI) support, creating a fully thermoset membrane structure. This design significantly enhances mechanical strength and mitigates irreversible compaction, which is a major source of performance degradation in standard thin-film composite (TFC) membranes.

The authors present a well-structured investigation into membrane fabrication, optimizing key parameters such as polymer concentration, solvent composition, and crosslinking method. They compare *ex situ* and *in situ* crosslinking approaches for the PI support, both of which yield robust structures suitable for polyamide layer formation via interfacial polymerization. Performance is further enhanced by incorporating triethylamine (TEA) in the polymerization step, producing tailored TFX membranes with significantly improved water permeance, high salt rejection (>99%), and strong compaction resistance under extreme salinity and pressure. The study is supported by comprehensive characterization, including FTIR, XPS, SEM, and *in operando* imaging, which confirms the structural integrity of TFX membranes compared to commercial HPRO counterparts.

The study appears to be solid and provides important insights to developing high-pressure compatible membranes. There are a few points that the authors may want to consider:

Our Response:

We sincerely thank the reviewer for the positive evaluation of our manuscript and for recognizing the novelty and significance of our TFX membrane platform. We appreciate your thoughtful summary of our design rationale, particularly the development of a fully crosslinked thermoset structure to mitigate irreversible compaction under ultrahigh pressures up to 200 bar. We are also grateful for your acknowledgment of the thoroughness of our membrane fabrication, optimization strategies, and comprehensive characterization using FTIR, XPS, SEM, and *in operando* imaging. We have carefully addressed each of the specific points raised in your subsequent comments, and we believe that the revisions have substantially improved the quality, clarity, and impact of the manuscript.

Comment 1:

L74-75, this claim on the universal benefit of compaction reduction across all applications has not been rigorously proven. We don't know exactly how much performance enhancement the proposed approach would bring to the low-pressure applications. Also, it has been argued that higher permeability membranes can only bring marginal improvement to the energy consumption and cost for SWRO.

Our Response:

We appreciate the reviewer’s insightful comment. To clarify, our intention was to emphasize the potential—rather than universal—benefits of integrating compaction-resistance features into a broad range of RO applications. We agree that the quantitative benefits, especially in low-pressure systems, are not yet well established and merit further investigation. To avoid any misunderstanding, we have revised the text accordingly:

Line 73 – 76: Potentially, state-of-the-art RO membranes could benefit from enhanced compaction resistance—including in low-pressure brackish water RO, ultralow pressure wastewater RO, and offshore sulfate removal nanofiltration (NF) membranes—by improving structural robustness, operational stability, and service lifespan.

Comment 2:

Fig. 3a, it appears that HPRO membrane is still superior even at 120 bar despite the compaction. The advantage of t-TFX is evident at 200bar. This point should be acknowledged. Also, the authors attribute the low permeability of TFX membranes (vs. HPRO or PSU-TFC) to the dense support layer. It’d be helpful to provide a breakdown of resistance between the active and support layer for the different membranes under different pressures.

Our Response:

Great points. We agree that the HPRO membrane demonstrates higher water permeance than the TFX membranes at 120 bar, and we have now acknowledged this in the revised texts. We also agree that the reduced permeability of TFX membranes is primarily due to the presence of a dense interfacial layer, rather than the bulk support layer alone. As reported in our recent study (Wu et al., 2025), most of the resistance is attributed to the active and interfacial layers, shown in the figure below. This understanding reinforces the importance of optimizing the interfacial region to further enhance the performance of TFX membranes without compromising compaction resistance.

[REDACTED]

We have revised the manuscripts to further clarify the relatively low water permeance of the TFX membranes is due to the tight interfacial layer, also to acknowledge the higher permeance of the commercial HPRO at 120 bar:

Line 308 – 313: At pressures below 120 bar, the commercial HPRO membrane displays higher water permeance than TFX membranes, attributable to its more porous interfacial and support layers (Fig. 1e–g). Most membrane resistance stems from the active layer and the interfacial layer in the composite membranes. Future work should prioritize enhancing the permeance of the interfacial layer of the support membranes while preserving compaction resistance¹.

[1] Wu, J. et al. Role of Transmembrane Pressure and Water Flux in Reverse Osmosis Composite Membrane Compaction and Performance. *Environmental Science & Technology* 2025. DOI: [10.1021/acs.est.5c02618](https://doi.org/10.1021/acs.est.5c02618).

Comment 3:

following the last two point, the authors should consider providing a high-level quantitative analysis regarding how the proposed membrane with reduced compaction can benefit high TDS brine concentration. E.g., <https://doi.org/10.1073/pnas.2022196118>

Our Response:

Great suggestion. We have now incorporated high-level quantitative discussion on how membrane with reduced compaction can benefit high TDS brine concentration:

Line 324 – 338: The improved structural resilience of TFX membranes under ultrahigh pressures directly enhances their applicability for high TDS brine concentration. Compaction-resistant membranes like *t*-TFX retain over 90% of their initial permeance ($A_{200}/A_{60} > 0.9$), while maintaining over 99% salt rejection, compared to commercial HPRO membranes that suffer ~70% loss in permeance and significant rejection decline under ultrahigh pressures. This performance enables single-stage concentration of brine beyond 180 g/L NaCl, reducing the need for complex multistage systems and associated capital costs². At 200 bar and 50% recovery, the specific energy consumption (SEC) of the TFX membrane system is ~9.7 kWh/m³, assuming 90% pump efficiency and 95% energy recovery efficiency¹. A recent study identified membrane water permeance as a top-value innovation target in high-salinity applications and showed that performance degradation—such as that from compaction—can drive sharp increases in LCOW unless mitigated by durable material designs². Thus, the thermoset architecture of TFX membranes not only ensures mechanical integrity but also delivers measurable energy and cost savings in high-recovery brine concentration processes.

[1] Anvari, A. *et al.* What will it take to get to 250,000 ppm brine concentration via ultra-high pressure reverse osmosis? And is it worth it? *Desalination* **580**, 117565 (2024). <https://doi.org/10.1016/j.desal.2024.117565>

[2] Dudchenko, A. V., Bartholomew, T. V. & Mauter, M. S. High-impact innovations for high-salinity membrane desalination. *Proceedings of the National Academy of Sciences* **118**, e2022196118 (2021). <https://doi.org/10.1073/pnas.2022196118>

Comment 4:

Fig. 3. The authors should consider reporting water flux, which goes with the observed rejection, as both are raw data, then A and B. In its current form it's not clear to the review that if the flux for HPRO and PSU-TFC goes up or down when pressure is increased. If the flux goes up for HPRO, with the near constant B value, rejection should go up as well. The authors should have also reported the concentration and osmotic pressure of the feed solution in the caption. Lastly, ob in Rob should be subscript.

Our Response:

We thank the reviewer for this helpful suggestion. In response, we have included the raw water flux data of all tested membranes in the supplementary information to accompany the reported A values. This addition improves clarity. In addition, we have corrected the formatting of R_{ob} to properly use subscript notation throughout the manuscript and figures for consistency. The testing conditions are reported in the manuscript.

Fig. S10 TFX membrane versus commercial HPRO and handcast PSU-TFC membrane wet-testing performance. Water permeance is presented in LMH/bar (liter $m^{-2} h^{-1} bar^{-1}$) and flux is presented in LMH ((liter $m^{-2} h^{-1}$)).

Line 504 – 506: In salt rejection tests, different concentrations of NaCl (Sigma Aldrich, S7653) were prepared as feed solution to maintain ~ 30 bar trans-membrane hydraulic pressure. For example, at 200 bar, the osmotic pressure of the feed solution was set at 170 bar.

Line 296 – 298: The slightly more compaction observed in t -TFX in contrast to e -TFX and i -TFX could be attributed to its higher flux (Supplementary Fig. S10)¹.

[1] Wu, J. et al. Role of Transmembrane Pressure and Water Flux in Reverse Osmosis Composite Membrane Compaction and Performance. Environmental Science & Technology 2025. DOI: [10.1021/acs.est.5c02618](https://doi.org/10.1021/acs.est.5c02618).

Comment 5:

the method section, especially the equations, is written with little care and full of minor errors. E.g., L444, J_{sw} should be J_s , same for L452, r_s and R_o are not consistent in capitalization, R_o here is R_{ob} in the figure, eqn 10 is missing a preceding factor that should include an osmotic coefficient evaluated using the Pfitzer model for high salinity. R_{PEG} and r_{PEG} are not consistent in eq.11 and 12. The use of dot and cross multiplication signs is very random. Font setting is very careless, inconsistent use of font size and type, the authors should learn from papers in math and physics.

Our Response:

We thank the reviewer for this detailed and constructive feedback. We have carefully revised the entire Methods section to ensure consistency and clarity in the equations, variable notations, and font formatting.

Reviewer 3

Comment 0:

In this original research work, the authors present an approach for the preparation of a new class of thin-film composite membranes with application in high-pressure reverse osmosis (HPRO). Membrane compaction is a well-known issue in RO technology and is currently regarded as one of the major limitations for achieving higher recovery in desalination and brine concentration. As described by the authors, membrane compaction is tightly connected to the thermoplastic nature of the support membrane (typically PSU-PET) and the presented approach relies on the transition from a thermoset PA coating film over a thermoplastic PSU-PET support membrane to a fully thermoset composite membrane (TFX).

Overall, the manuscript is very well written and organized with a clear line of argument. The presented TFX membranes exhibit high compaction resistance and maintain a salt retention of over 99 % for NaCl up to 180 g/L as well as a satisfying water permeance at a pressure of 200 bar. Advanced imaging techniques show that, in contrast to commercial TFC-HPRO membranes, the TFX membranes deform significantly less during and after compression up to 200 bar. This finding is further supported by the measurement of pore size distributions before and after compaction. It is worth mentioning that the presented approach can be scaled up by a moderate upgrade of the current reverse osmosis membrane production technology. With the help of systematic optimization, the authors provide a toolbox for fine-tuning both the uncoated and the coated membrane. The in-depth analysis of the membrane performance suggest an interesting separation behavior of the TFX during compaction. The rejection of some membrane samples was improved during compaction.

The key point of this work relies in the interesting and very convincing approach to compaction resistant RO membranes that should be optimized for ultra-high pressure operations. Conventional development approaches aim to maintain membrane performance over a wide pressure range. Here the authors suggest to focus the optimization efforts on the particular conditions of the intended application namely ultra-high pressure. It is not about maintaining the performance but playing with the membrane dynamics to reach optimal performance at the desired pressure. The methodology is sound and presented in detail so that the experimental work can be reproduced. I could not find any flaws in the data analysis, interpretation and conclusions.

The excellent compaction resistance of the resulting TFX membranes up to 200 bar (<10% decline in water permeance) is particularly noteworthy. With a maximal operating pressure of 120 bar, Dupont XUS 1808 polyamide composite reverse RO membrane modules currently set the benchmark in HPRO operations. From this perspective, a shift in maximal operating pressure from 120 bar to 200 bar represents a very significant step in this field. The impact of this research goes beyond desalination and brine concentration. Indeed, such a progress in compaction resistance opens up new perspectives for HPRO, particularly in the areas of resource recovery & reuse from industrial wastewater as well as zero liquid discharge (ZLD). High-pressure batch RO processes that enable the simultaneous recovery/reuse of resources and production of clean water are well suited to closing loops, limiting liquid waste and preserving fresh water. The reutilization of recovered resources

often requires a considerable concentration of brines, which leads to high osmotic and operating pressures. Such a batch RO process was recently successfully developed for the recovery and reuse of Cr(III) from electroplating wastewater with a specific energy consumption (SEC) of approx. 2.5 kWh/m³ (10.1016/j.desal.2024.117479 and 10.1021/acsestwater.4c00556). It is operated at pressures up to 120 bar and involves Dupont XUS 1808 RO membrane modules. In this case, process efficiency is clearly limited by the compaction resistance of the membrane and substantial improvement could be achieved by operating the system at pressure > 120 bar.

The authors presents an extensive dataset that reveals convincing evidence and strongly supports their claims and conclusions. The dataset combines independent characterization techniques ranging from advanced imaging techniques to the full determination of the properties and the performance (dry and wet-testing) of uncoated and coated membranes up to in operando characterization and quantitative evaluation of the composite membranes before and after compaction. The commercial Dupont TFC-HPRO membrane is the reference of choice for this work and the A200/A60 compaction resistance indicator is very well appropriate to demonstrate the gain in mechanical stability during the optimization the composite membranes for ultra-high pressure applications.

Overall, this is an impressive work and there is, from my side, very little to criticize.

Our Response:

We deeply appreciate your thorough evaluation and positive comments on our manuscript. We are pleased that the reviewer recognizes the significance of our work on developing thin-film crosslinked composite membranes (TFX) for ultra-high-pressure reverse osmosis (UHPRO) applications.

We have addressed all of the reviewer's suggestions and corrected the typographical errors highlighted. Additionally, we have expanded our manuscript's discussion regarding the potential applications of HPRO and UHPRO membranes as recommended by the reviewer. Specifically, we added relevant references to highlight the importance of compaction-resistant membranes for various critical applications, as follows:

Line 70 – 73: Developing compaction-resistant membranes is particularly important for high pressure and ultrahigh pressure RO applications such as seawater desalination, direct brine concentration, brine concentration via osmosis-assisted multi-stage RO processes, and various resource recovery operations¹⁻³.

[1] Anvari, A. *et al.* What will it take to get to 250,000 ppm brine concentration via ultra-high pressure reverse osmosis? And is it worth it? *Desalination* **580**, 117565 (2024).

[2] Karimi, S. *et al.* High-pressure batch reverse osmosis (RO) for zero liquid discharge (ZLD) in a Cr (III) electroplating process. *Desalination* **580**, 117479 (2024).

[3] Engstler, R. *et al.* A Robust High-Pressure RO Technology to Overcome the Barriers to Full Circularity in Cr (III) Electroplating Operations. *ACS Es&t Water* **4**, 5461-5472 (2024).

Comment 1:

I think there is a typo regarding the reference Dupont TFC-HPRO membrane that is referred as Dupont XUS 1818. I am not aware of this membrane type. To the best of my knowledge, the HPRO membranes is referred to as Dupont XUS 1808.

Our Response:

Thanks for your comment. We have revised the typo and it should be Dupont XUS 1808:

Line 490 – 491: A commercially-available, TFC-HPRO membrane (Dupont XUS1808, HPRO) was tested along with hand-cast PI-TFX, PI-TFC and PSU-TFC membranes in this work.

Comment 2:

It would be helpful to provide a picture showing the fabrication flow of the most promising membrane and highlighting that the current state of the art can be adapted to host the new technology.

Our Response:

We thank the reviewer for this insightful suggestion. We fully agree that a schematic clearly illustrating the transition from conventional TFC membrane manufacturing processes to TFX membrane production would significantly enhance the manuscript. Accordingly, we have added a new schematic (Fig. 3f) demonstrating how existing commercial TFC manufacturing lines can be efficiently adapted for roll-to-roll production of TFX membranes.

Fig. 3f. Schematic illustration of adapting existing TFC membrane manufacturing lines for roll-to-roll production of TFX membranes.

Line 350 – 362: To enable large-scale production of TFX membranes, existing TFC membrane manufacturing lines can be strategically adapted with minimal modification. As illustrated in Fig. 3f, the conventional TFC production process—which typically involves casting a polymer solution onto a nonwoven fabric followed by coagulation in a non-solvent bath—can be seamlessly integrated with in-line chemical crosslinking of the support layer. After the formation of the crosslinked support, the membrane passes through subsequent interfacial polymerization steps involving immersion in aqueous amine and organic acyl halide solutions, followed by thermal curing in an oven and post-treatment. This modular adjustment to the TFC roll-to-roll process enables the formation of a fully thermoset composite structure, characteristic of TFX membranes, without disrupting the overall production flow. By leveraging the existing industrial infrastructure, this approach facilitates rapid translation of TFX technology from lab-scale to scalable, industrial membrane fabrication.

Comment 3:

A few typos:

Line 77: the article the is repeated

Line 81: as an example of polymer or as an exemplary polymer

Line 98: systematic optimization.... This sentence has no verb.

Our Response:

Thank you for carefully noting these typographical and grammatical issues. We have revised the manuscript accordingly.

Reviewer 1

Comment 0:

The manuscript has been significantly revised to improve clarity by addressing most of the comments. However, there are still some minor concerns to address.

Our Response:

We sincerely appreciate the reviewer's recognition of our efforts to improve the manuscript. To further enhance clarity and address the remaining concerns, we have made additional revisions, which include:

1. Expanding the description of the *in operando* SEM imaging technique (Fig. S11);
2. Providing more detailed comparisons of the structural and morphological differences between support membranes formed via *in situ* versus *ex situ* crosslinking;
3. Clarifying how these crosslinking procedures affect the support membrane's surface and bulk porosity, thereby influencing monomer uptake and ultimately governing the structure and performance of the polyamide selective layer.

We believe these revisions have further strengthened the manuscript, and we respectfully consider it ready for publication.

Comment 1:

The authors have added the detail experimental procedure for the *in operando* SEM. However, a schematic or photo of the *in operando* SEM setup (instrument) under high pressure should be included either in the main manuscript or in the SI, as this nonconventional technique is only referenced via a PhD thesis that cannot be found online through the university repository search. A clear image would aid reader understanding, particularly for non-specialists.

Our Response:

Thanks for the suggestion. We agree that the schematic or photo of the *in operando* SEM methodology is helpful. We have now incorporated the schematic and photos of *in operando* SEM imaging set-up to SI.

Line 579 – 587:

In operando SEM characterization and AI segmentation of the composite membranes under pressure were performed using a CrossBeam 340 SEM (Carl Zeiss Microscopy, LLC, NY, USA) equipped with an *in operando* compression stage. The schematic of the *in operando* SEM imaging setup is included in Fig. S11. Membrane coupons with a clean cross-sectional area of at least $20,000\ \mu\text{m} \times 50\ \mu\text{m}$ were carefully prepared and mounted within the compression chamber. Compression was applied perpendicular to the membrane surface, with pressure gradually increased to 30, 60, and 120 bar. High-resolution SEM images were recorded at each pressure level to visualize morphological changes under realistic operating conditions.

Fig. S11 *In operando* SEM imaging setup. The Zaber actuator provides precise control over the movement of the compression stage. The phidgets load cell measures the force applied to the sample with high accuracy. The phidgets bridge board connects the load cell to the control system, enabling signal processing and data acquisition. The electron beam (E-beam) enables real-time imaging. The *in situ* compression stage is customized for evaluating membrane compaction behavior.

Comment 2:

Comment 7 remains partially unaddressed. The difference in permeance between *in situ* and *ex situ* treatments is more pronounced than the difference in rejection. For instance, 1:2-*ex situ* and 1:2-*in situ* shows permeance values of 20.6 and 16.4 LMH/bar, respectively (a 26% decrease), while their rejection rates remain nearly identical at 96.7% and 97.2%. It is worth considering that this likely stems from differences in surface and bulk effects: *in situ* treatment may influence both, leading to uniform densification, whereas *ex situ* primarily affects the surface, maintaining rejection performance but altering permeance.

Our Response:

Thank you for the insightful comment. We agree and have now incorporated a more detailed discussion of the effects of crosslinking approach into the revised manuscript:

Line 165 – 175:

From a structural standpoint, both *in situ* and *ex situ* crosslinking reduce surface porosity by promoting polymer densification, but their crosslinking profiles may differ due to diffusion kinetics. *Ex situ* crosslinking typically results in more surface-concentrated modification, while *in situ* crosslinking enables more uniform integration as the diamine diffuses through the semi-coagulated polymer during phase inversion. This difference is evident in membranes with more porous structures, such as in the comparisons of 1:2 *ex situ* vs. 1:2 *in situ* and 1:1 *ex situ* vs. 1:1 *in situ*. In these cases, the *in situ* approach leads to more extensive densification throughout the membrane matrix, resulting in lower water permeance and slightly higher rejection. In contrast, the *ex situ* method primarily modifies the surface layer, maintaining rejection performance but allowing higher water permeance due to less uniform densification.

Comment 3:

The response to comment 10 needs more detail. The authors suggest that differences in crosslinking procedures (*ex situ* vs. *in situ*) affect surface morphology due to variations in monomer uptake and polyamide layer formation. However, they do not explain how these processes (monomer uptake and polyamide layer formation) occur in each method.

Our Response:

Thank you for the insightful comment. We have revised the manuscript to more clearly articulate how differences in crosslinking procedures (*in situ* vs. *ex situ*) influence the structure of the support membrane and, in turn, affect monomer uptake and polyamide (PA) layer formation:

Line 218 – 238:

Although all TFX membranes were coated using the same formulation, the resulting active layer morphologies differed (Fig. S7), re-confirming how subtle differences in support properties can affect monomer uptake and polyamide layer development¹. For *in situ* crosslinking, the crosslinker is present during phase inversion, allowing it to diffuse into the nascent, semi-coagulated polymer matrix. This facilitates more uniform densification throughout the support membrane's near surface layer, reducing both surface and body porosity. As established in previous study¹, reduced bulk porosity directly limits the uptake of the aqueous MPD monomer, which subsequently lowers the local amine concentration available during interfacial polymerization. This leads to the formation of thinner PA layers with reduced mass and crosslinking degree, thus increasing water permeance, but potentially reducing rejection.

In contrast, *ex situ* crosslinking occurs after the support membrane has already formed. The crosslinking reaction primarily affects the surface region. As a result, *ex situ* treated supports typically retain higher body porosity, enabling greater MPD uptake, especially near the top surface. This increased monomer availability translates into thicker and more crosslinked PA layers, leading to lower water permeance and higher salt rejection. Therefore, the variation in PA layer structure between *in situ* and *ex situ* crosslinked supports arises from differences in both surface and bulk porosity, which control the mass of monomer sequestered within the support. This, in turn, determines the MPD:TMC ratio in the interfacial reaction zone and governs the final PA layer morphology and performance.

(1) Dlamini, D. S.; Quezada-Renteria, J. A.; Wu, J.; Xiao, M.; Anderson, M.; Kaner, R. B.; Edalat, A.; Voutchkov, N.; Al-Ahmoudi, A.; Hoek, E. M. On the role of the porous support membrane in seawater reverse osmosis membrane synthesis, properties and performance. *Journal of Membrane Science* **2024**, *708*, 123032.

Comment 4:

In the author contribution section, the authors did not mention the contribution of all authors. Moreover, in this revised version, a new author is added. Justify the need for adding new authors and detail the contribution of each author.

Our Response:

Thank you for the helpful comment. The newly added author, Fiona Chen, was involved in conducting atomic force microscopy (AFM) imaging and contributed significantly to the analysis of those results, which were newly included in the revised manuscript. Based on this meaningful contribution to data generation and interpretation, her authorship is warranted. We have updated the author contribution section as follows:

Author contributions: J.W. and E.H. conceptualized and designed the study. J.W., J.Q., J.L., K.A., K.G., and T.T. conducted experimental research. J.W., M.X., and F.C. performed imaging. J.W., Y.C., H.F., X.W. and J.Q. performed data analysis and visualization. All authors discussed results. E.H. supervised the study. J.W. wrote the manuscript, with all authors contributing to manuscript editing.

Reviewer 2

Comment:

The authors have satisfactorily addressed all my concerns. the manuscript is ready for publication.

Our Response:

We sincerely thank the reviewer for the positive feedback and are grateful for your time and thoughtful evaluation throughout the review process.

Reviewer 3

Comment:

Thank you very much for the thoughtfully revision of the manuscript, which can be published in the present form. Well done!

Our Response:

We sincerely thank the reviewer for the encouraging feedback and are grateful for your positive evaluation of our revised manuscript.